# Cross-platform normalization enables machine learning model training on microarray and RNA-seq data simultaneously

Steven M. Foltz [1,2], Casey S. Greene [1,3,4✉] & Jaclyn N. Taroni [1,2✉]

Large compendia of gene expression data have proven valuable for the discovery of novel biological relationships. Historically, most available RNA assays were run on microarray, while RNA-seq is now the platform of choice for many new experiments. The data structure and distributions between the platforms differ, making it challenging to combine them directly. Here we perform supervised and unsupervised machine learning evaluations to assess which existing normalization methods are best suited for combining microarray and RNA-seq data. We find that quantile and Training Distribution Matching normalization allow for supervised and unsupervised model training on microarray and RNA-seq data simultaneously. Nonparanormal normalization and z-scores are also appropriate for some applications, including pathway analysis with Pathway-Level Information Extractor (PLIER). We demonstrate that it is possible to perform effective cross-platform normalization using existing methods to combine microarray and RNA-seq data for machine learning applications.

[1] Department of Systems Pharmacology and Translational Therapeutics, Perelman School of Medicine, University of Pennsylvania, Philadelphia, PA, USA.
[2] Childhood Cancer Data Lab, Alex's Lemonade Stand Foundation, Wynnewood, PA, USA. [3] Center for Health AI, University of Colorado School of Medicine, Aurora, CO, USA. [4] Department of Biomedical Informatics, University of Colorado School of Medicine, Aurora, CO, USA. ✉email: casey.s.greene@cuanschutz.edu; jaclyn.taroni@ccdatalab.org

The union of large and diverse compendia of gene expression data with machine learning approaches has enabled the extraction of cell type-specific networks[1] and the discovery of new biological patterns associated with cellular responses to the environment[2]. Integrative analyses of multiple microarray cohorts have uncovered important signatures in human infection[3,4]. Sequencing-based RNA assays have certain advantages over array-based methods, namely quantitative expression levels, a lack of dependence on current annotations, and a higher dynamic range[5]. As a result, researchers have increasingly adopted this technology for their gene expression experiments.

RNA-sequencing (RNA-seq) assays represent a growing share of new gene expression experiments. In February 2022, the ratio of summarized human microarray to RNA-seq samples from GEO and ArrayExpress was close to one to one (1.13:1), meaning that half of all summarized human samples come from either platform, although RNA-seq overtook microarray data as the leading source of new submissions to ArrayExpress in 2018[6–9]. Integrative analyses of gene expression requires the combination of these data types—a task of utmost importance for rare diseases or understudied biological processes and organisms where all available assays will be required to discover robust signatures or biomarkers. Thus, effective strategies for combining data from the two platforms—perhaps in a manner that leverages both the advantages of RNA-seq and the abundance of microarray data—are paramount to transcriptomic and functional genomic experiments going forward.

Much work has been performed to develop methods for effectively combining multiple cohorts or batches of gene expression data, but no method has been widely adopted for the problem of combining mixtures of array and RNA-seq platform data[4,10–16]. Quantile normalization (QN) is a widely used normalization technique originally utilized for microarray data[17], and it has also been adopted for RNA-seq data normalization[18] and in some cases, cross-platform normalization[19]. Probe Region Expression estimation Based on Sequencing (PREBS) was developed to make RNA-seq and microarray data more comparable, but this method requires raw reads and probe specific information and may not be feasible for large-scale public data efforts[20]. Training Distribution Matching (TDM) was developed by our group to make RNA-seq data more comparable to microarray data from transcript abundances specifically for machine learning applications[21]. In that work, it was demonstrated that QN, TDM and a method from the analysis of graphs, nonparanormal normalization (NPN), had good performance in the supervised learning evaluation. However, combining array and RNA-seq platforms was not evaluated. Recent efforts to integrate single-cell RNA-seq data across experiments and modalities has generated many new approaches[22,23].

Here, we present a series of experiments to test what normalization approaches can be used to combine microarray and RNA-seq data for supervised machine learning and unsupervised machine learning, including pathway analysis. We chose subtype and mutation status prediction for our supervised machine learning tasks because they are commonly used in cancer genomics studies and the labels are well-defined in our data. Our unsupervised machine learning results focus on pathway analysis since this is an important downstream application for understanding biologically relevant gene expression patterns. Given the broader dynamic range of RNA-seq data, this work focuses on the problem of normalizing RNA-seq data to a target distribution of array data. We specifically add varying numbers of RNA-seq data to our training sets to assess at what point performance begins to suffer. We find that QN, TDM, NPN, and standardized scores are all suitable for some use cases, with the widely adopted QN performing well for machine learning applications in particular.

## Results

We performed a series of supervised and unsupervised machine learning evaluations to assess which normalization methods are best suited for combining data from microarray and RNA-seq platforms. We evaluated seven normalization approaches for all methods: LOG, NPN, QN, QN (CN), QN-Z, TDM, and standardizing scores (z-scoring; Z), plus untransformed data for comparison (UN) added as an additional negative control.

**Non-paranormal normalization, quantile normalization, and training distribution matching allow for training subtype and mutation classifiers on mixed platform sets**. We trained models to predict subtype from our BRCA and GBM training sets with varying numbers of samples from the RNA-seq platform. We used these models to predict subtype on holdout data sets composed entirely of microarray data or RNA-seq data (Fig. 1). We trained three commonly used classifiers: LASSO logistic regression, linear SVM, and random forest. Kappa statistics were used to assess performance, in addition to AUC, sensitivity, and specificity (Supplementary Data 1). Although each metric showed similar results, we focused on Kappa statistics for our interpretation of model performance to directly incorporate the multi-class and imbalanced nature of our data sets. We visualize the BRCA Kappa statistics for varying numbers of samples from RNA-seq in the training data in Fig. 2 and show GBM results in Supplementary Fig. 3. (Note that the pipeline in Fig. 1a–c was repeated ten times.) The three classifiers showed the same trends across normalization approaches overall, suggesting that the normalization approaches recommended herein will generalize to multiple classification methods.

Importantly, for BRCA, there were appreciable differences between normalization methods (Fig. 2). GBM results showed similar results but with less distinction between methods (Supplementary Fig. 3) and lower Kappa values than BRCA overall. Such differences may come from the type of RNA-seq data used (RSEM counts for BRCA, UQ-FPKM for GBM), the size of training data sets used (BRCA 348 samples, GBM 102 samples), or how well each cancer's subtypes are defined by gene expression. Log-transformation demonstrated among the worst performance. This is expected as we consider this method to be a negative control and it was previously shown to be insufficient to make RNA-seq data comparable to microarray[21]. We also saw that z-scoring data resulted in the most variable performance. This is not unexpected because the calculation of the standard deviation and mean will be highly dependent on which samples are selected from each platform and the random selection of RNA-seq samples to be included in the training set does not consider subtype distribution, which may not be known in practice. We found that NPN, QN, QN-Z, and TDM all performed well when moderate (i.e., not extreme) amounts of RNA-seq data were incorporated into the training set. These results are consistent with the high performance of these three methods in our case study of training entirely on microarray data and using solely RNA-seq as a test set[21]. These three methods performed well on both the microarray and RNA-seq holdout sets. QN followed by z-scoring (QN-Z) showed similar results to QN without z-scoring. Untransformed data (UN) showed poor performance at all titration levels.

Quantile normalization did not perform well at the extremes (0 and 100% RNA-seq data). We attribute this loss of performance to the lack of reference distributions in these cases—for all other amounts of RNA-seq (10–90%), the set of

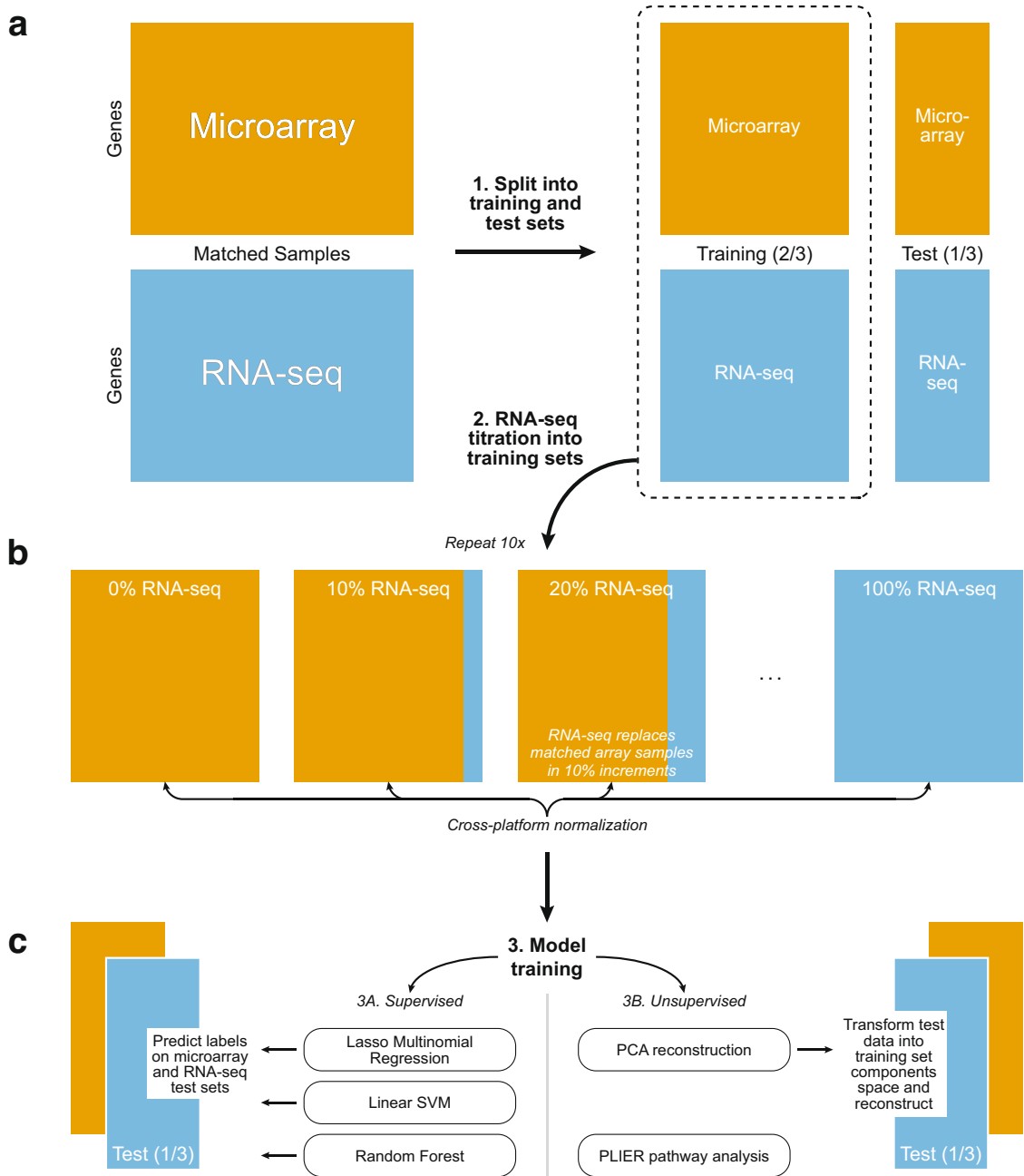

**Fig. 1 Overview of supervised and unsupervised machine learning experiments. a** TCGA matched samples (520 from BRCA, 150 from GBM) run on both microarray and RNA-seq were split into a training set (2/3) and test set (1/3). **b** RNA-seq samples were titrated into each training set, 10% at a time (0–100%), resulting in eleven training sets for each normalization method. Each RNA-seq sample replaces its matched microarray sample. Cross-platform normalization methods were applied to each training set independently. **c** We used three supervised algorithms to train classifiers (molecular subtype and mutation status of *TP53* and *PIK3CA* in both BRCA and GBM) on each training set and tested on the microarray and RNA-seq test sets. The test sets were projected onto and back out of the training set space using unsupervised Principal Components Analysis to obtain reconstructed test sets. The subtype classifiers trained in step 3A were used to predict on the reconstructed test sets. Pathways regulating gene expression were identified using the unsupervised method PLIER.

microarray data is used as a reference distribution for both the RNA-seq data included in the training set as well as for the holdout set (see Methods and Data). This result reiterates the importance of drawing training and holdout sets from the same distribution, as is well-documented in the machine learning literature, and highlights the necessity of proper cross-platform normalization.

We also predicted *TP53* and *PIK3CA* mutation status in BRCA and GBM. We report results for *TP53* prediction in GBM in

Fig. 3 (others in Supplementary Figs. 4–5), using delta Kappa, the difference in Kappa between models with true labels and null models with mutation labels randomized within subtype. This was necessary because mutation class imbalance differs between molecular subtypes. Delta Kappa values close to zero indicate little improvement of the true model over the null model. Although delta Kappa values were mostly positive, indicating an improvement of the true label models over the null models, there was little difference in performance between normalization

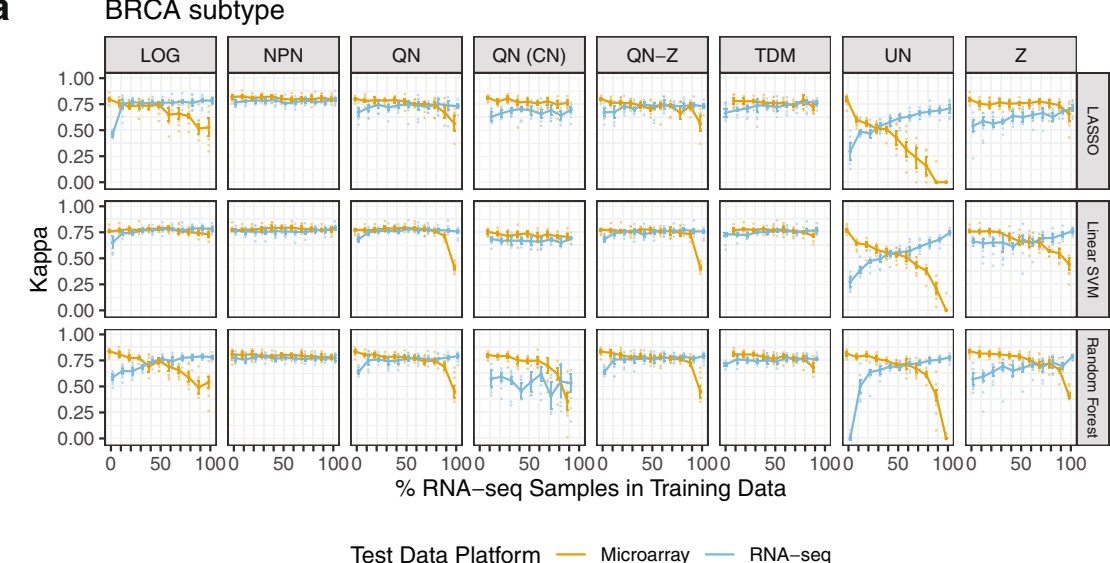

**Fig. 2 BRCA subtype classifier performance on microarray and RNA-seq test data. a** Median Kappa statistics from 10 repeats of steps 1–3 A from Fig. 1 and for seven normalization methods (and untransformed data) are displayed. Median values are shown as points, and approximate 95% confidence intervals are shown around each median defined as $+/-1.58*IQR/sqrt(n)$ with IQR = interquartile range and $n$ = number of observations[60]. LOG log2-transformed, NPN nonparanormal normalization, QN quantile normalization, QN (CN) quantile normalization with CrossNorm, QN-Z quantile normalization followed by z-score, TDM Training distribution matching, UN untransformed, Z z-score.

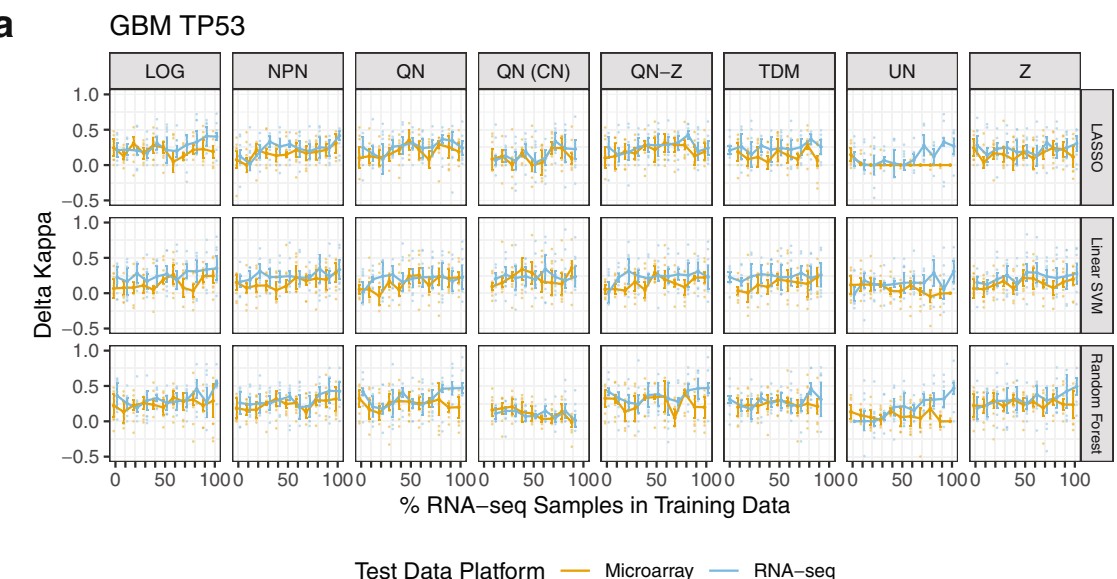

**Fig. 3 GBM *TP53* mutation classifier performance on microarray and RNA-seq test data. a** Median delta Kappa statistics from 10 repeats of steps 1–3 A from Fig. 1 and for seven normalization methods (and untransformed data) are displayed. Median values are shown as points, and approximate 95% confidence intervals are shown around each median defined as $+/-1.58*IQR/sqrt(n)$ with IQR = interquartile range and $n$ = number of observations[60]. Delta Kappa measures the difference in Kappa values between a null model and a model built with true labels. LOG log2-transformed, NPN nonparanormal normalization, QN quantile normalization, QN (CN) quantile normalization with CrossNorm, QN-Z quantile normalization followed by z-score, TDM training distribution matching, UN untransformed, Z z-score.

methods or across RNA-seq titration levels among LOG, NPN, QN, QN-Z, and TDM. Z and UN results varied widely across the spectrum of RNA-seq titration level. *PIK3CA* mutation prediction in GBM showed negligible improvement over the null model, possibly due to the low proportion of samples with mutations ($n = 10$ out of 98 samples in training) (Supplementary Fig. 5b).

**Suitable normalization methods for unsupervised learning depend on the downstream application**

*Downstream analysis of pathways regulating gene expression.* We ran PLIER[24] to identify pathways significantly associated with at least one latent variable in gene expression data derived from a single platform (microarray only or RNA-seq only) or mixed-platform (combination of microarray and RNA-seq). Here, we

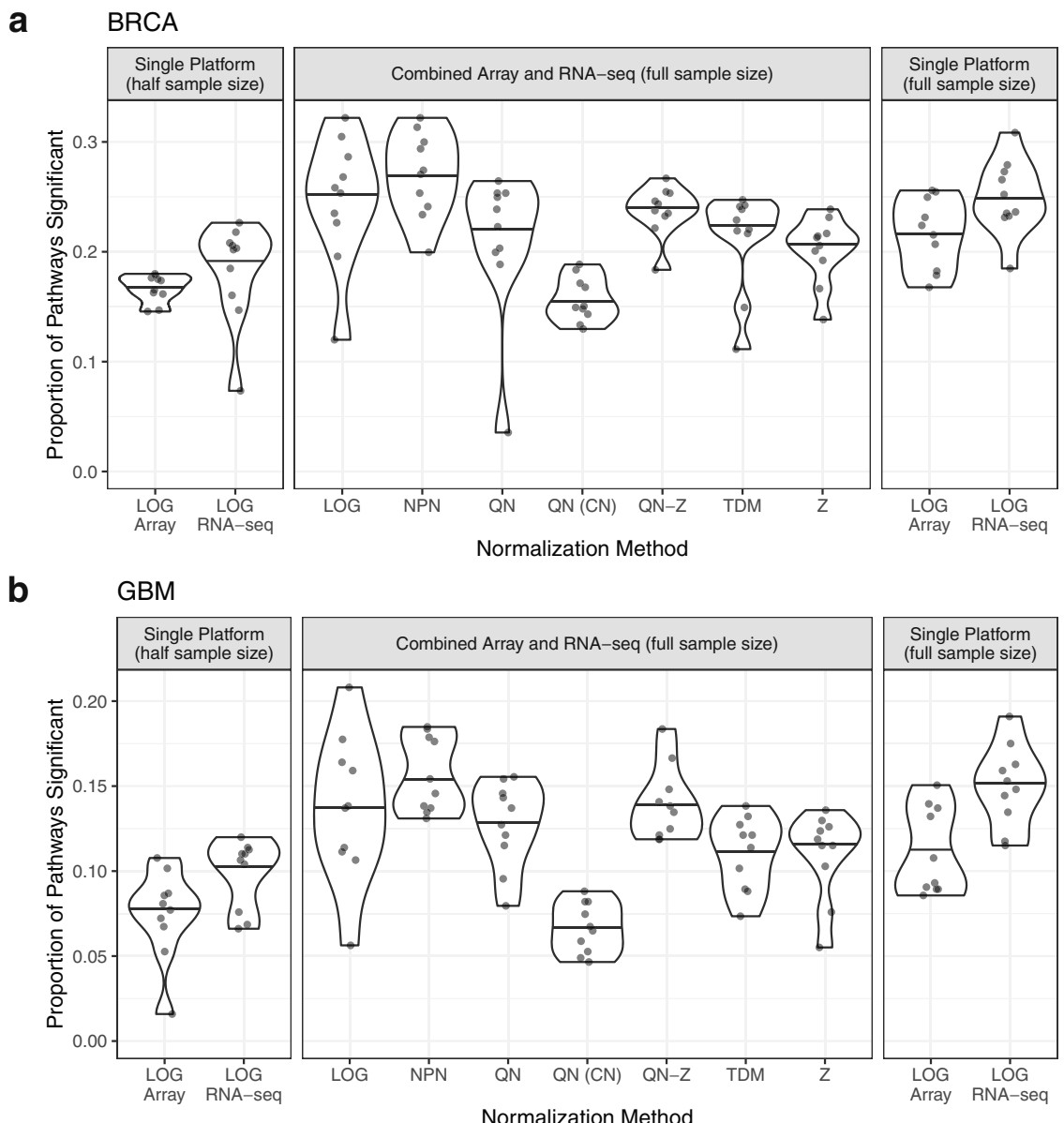

**Fig. 4 Pathways regulating gene expression identified by PLIER.** Pathway-level information extractor (PLIER) analysis results showing the proportion of pathways associated with at least one latent variable for BRCA (**a**) and GBM (**b**). The Single Platform (half sample size) panel shows single platform data using half the available samples for each platform. The Combined Array and RNA-seq panel shows 50% array and 50% RNA-seq data combined and normalized by various methods representing the full set of available samples. The Single Platform (full sample size) panel shows single platform data using the full set of available samples for each platform. LOG log2-transformed, NPN nonparanormal normalization, QN quantile normalization, QN (CN) quantile normalization with CrossNorm, QN-Z quantile normalization followed by z-scoring, TDM training distribution matching, Z z-scoring.

wanted to examine the benefit of creating a larger data set through the combination of platforms for downstream analysis. For BRCA, the half-size single platform data comprised 174 samples (half of available training samples), while the full-size single platform and cross-platform data each had 348 samples (the full set of training samples). For GBM, the sample sizes were 51 for half-size and 102 for full-size data sets. As a negative control, we permuted gene-pathway relationships used as input to PLIER to establish the baseline rate of false positive pathway associations. We found that out of 110 PLIER runs, five runs reported one false positive significant pathway association, while 105 runs reported no significant pathways.

Overall, in both cancer types, doubling the sample size of our data sets resulted in a greater proportion of available pathways being significantly associated with an underlying latent variable in

the decomposed data set, with a similar increase observed for both cross-platform and single platform full-size data sets. This could indicate a greater biologically relevant signal being extracted from the data (Fig. 4a, b) and is consistent with previous studies[25]. We found that doubling the sample size led to important cancer pathways being identified more stably and regularly in both the single-platform and cross-platform settings, including pathways related to ERB2, NFKB immune infiltration, and RAF in BRCA, and KRAS, MYC, and PRC2-related pathways in GBM. Pathway results are included for BRCA (Supplementary Data 2) and GBM (Supplementary Data 3). For GBM, with its smaller sample size overall, the proportional benefit of doubling the number of samples was greater than the benefit seen in BRCA. Combined platform data normalized by NPN showed the highest proportion of significant pathways, suggesting that input

data closest to a normal distribution on a gene level may be best suited for PLIER. Untransformed data (UN), with its widely differing ranges for array and RNA-seq data, could not be successfully run with PLIER.

*Dimensionality reduction and reconstruction.* Dimensionality reduction and/or unsupervised feature extraction methods are commonly employed in the analysis of gene expression data. We used Principal Components Analysis (PCA) and evaluated normalization method performance. The molecular subtypes in BRCA and GBM are strong, linear signals that we should be able to predict in the holdout sets given the performance of the classifiers visualized in Fig. 2 and Supplementary Fig. 3 and should be readily extractable using PCA. We aimed to identify which normalization methods were most suitable for feature extraction in data sets comprising a mixture of microarray and RNA-seq data. Our approach was as follows: PCA was performed on the training sets and then the holdout sets were projected onto the training space and then reconstructed to obtain reconstructed holdout sets (see Methods, Fig. 1). We evaluated performance in two ways: (1) we performed BRCA and GBM subtype prediction on the reconstructed sets using the classifiers trained in the supervised analyses and (2) we calculated reconstruction error post-transformation (MASE; see Methods).

We observed differences in classifiers and normalization methods as measured by the Kappa statistics (Supplementary Figs. 6–7). For BRCA, the SVM performance was most robust to reconstruction (Supplementary Fig. 6), consistent with expectations for linearly separable class problems. In general, the random forest classifier suffered the largest loss of performance, likely due to gene expression thresholds (rules) used for prediction (Supplementary Fig. 7). In the case of NPN, the near zero random forest Kappa statistics (Supplementary Fig. 6) resulted from predictions of only one class label. We observed the largest differences in performance between the two platform holdout sets with Z normalization (Supplementary Fig. 6). We found that projecting the holdout sets onto QN and TDM normalized training space results in less loss of subtype classifier performance relative to when there is no reconstruction performed (Supplementary Fig. 7). In addition, we observed QN resulted in low reconstruction error (Supplementary Fig. 8). This suggests that QN and TDM are suitable for normalizing sets composed of data from both platforms for use with unsupervised feature extraction applications.

## Discussion

We examined cross-platform normalization methods for machine learning on training sets composed of data measured on RNA-seq and microarray, and we demonstrated that it is possible to combine these data types for use with supervised and unsupervised applications. In general, performance on holdout data from both platforms is largely comparable. We find that QN and TDM, when using moderate numbers of RNA-seq samples (10–90%), perform well for both types of machine learning approaches. NPN is a strong performer in the supervised ML applications and the unsupervised PLIER analysis but may not be appropriate in the unsupervised PCA reconstruction applications given the decrease in Kappa in predictions on reconstructed expression data. Other applications of cross-platform normalization, such as combining array-based gene expression data from different array manufacturers (e.g., Agilent and Affymetrix) could follow the same principles as what we have described here.

This study has some important limitations. Because of our experimental design, we required large data sets of samples run on both platforms. This focused our work on two high quality data sets: TCGA BRCA and GBM data. BRCA and GBM subtypes and common gene mutations are well-defined signatures that have evident linear expression patterns. As a result, our guidance may not generalize to nonlinear classifiers, data sets of poor quality, or small sample sizes. In the context of mutation status prediction, we found the mutation class balance to differ between subtypes. Therefore, the overall expected accuracy calculated for the Kappa statistic may not adequately reflect the expected accuracy if, for instance, the classifier predicts the majority mutation class within a subtype. This suggests that the Kappa statistic may be less well-suited for this evaluation compared to predicting molecular subtype.

In addition, we stress that the biological question at hand when performing cross-platform normalization must be considered, as some assumptions underlying a normalization method may be violated. By harmonizing RNA-seq to microarray data, we gain the benefit of larger sample sizes required for more data-intensive machine learning applications. But, while gene expression distributions can be adjusted, the properties of count-based data, especially with respect to ties and true zero values, may hinder our ability to measure performance across multiple tasks. Alternative approaches may not require reshaping RNA-seq data to match array data. For example, methods utilizing gene-pair ratios may rely on relative expression levels of genes within samples to identify useful features[26] and should be considered for future work, including recent approaches like meGPS that apply this idea across platforms and data types[27].

We designed these experiments to evaluate the combination of microarray and RNA-seq data in practice in the absence of samples measured on both platforms. This work indicates that it is possible to perform model training on microarray and RNA-seq. Combining both platforms could allow models to take advantage of the additional information captured in some RNA-seq experiments while benefiting from the historical abundance of microarray data. For example, with low false positive rates, we showed that increasing sample size with cross-platform normalization yielded more significant pathways in our PLIER analysis. Training sets comprising samples run on both platforms will be a new reality as RNA-seq becomes the platform of choice and the ability to perform such analyses will be of particular importance for understudied biological problems.

## Methods

We aimed to assess the extent to which it was possible to effectively normalize and combine microarray and RNA-seq data with existing methods for use as a training set for machine learning applications. We assessed performance on holdout sets composed entirely of microarray data and entirely of RNA-seq data. To design such an experiment, we required data sets that had matched samples—sets of samples run on both microarray and RNA-seq—and that were of sufficient size. Our matched samples design allowed us to directly evaluate how the addition of RNA-seq data into the training set affects performance without the added complication of a change in the underlying samples (e.g., biological replicates). Using a different set of samples for training or evaluation could have resulted in a change in performance that was unrelated to the platform difference, which could confound the results. For our machine learning experiments, we used both microarray and RNA-seq holdout sets composed of the same samples to assess how the addition of RNA-seq data into the training set affected test accuracy on both widely used genome-wide expression platforms.

Although our experimental design required matched samples, we designed an analysis to evaluate normalization methods for the purpose of combining RNA-seq and microarray data when matched samples are not available. We compare holdout performance with both platforms separately to assess the extent to which models trained on a mix of RNA-seq and microarray data performed differently on test sets from the two platforms (see Experimental design). We also took care to design the normalization process such that it is applicable for data sets without matched samples—that is, when reference distributions are required, we do not use all microarray samples to normalize the RNA-seq, but rather the subset of microarray data included in the training set (see Cross-platform normalization approaches).

**Evaluation gene expression and mutation data**. The Cancer Genome Atlas (TCGA) breast cancer (BRCA) and glioblastoma (GBM) data sets include samples that have been measured with both microarray and RNA-seq platforms[28–31]. In addition, both BRCA and GBM have well-defined molecular subtypes that are suitable for use as labels/classes for supervised machine learning approaches we describe below. BRCA and GBM also show recurrent *TP53* and *PIK3CA* gene mutations, allowing for the prediction of mutation status based on gene expression profiles[32–34]. For BRCA (520 pairs of matched samples), we used log₂-transformed, lowess normalized Agilent 244 K microarray data[29] and RSEM (RNA-seq by Expectation Maximization) gene-level count RNA-seq data[35]. For GBM (150 pairs of matched samples) we obtained Affymetrix HT Human Genome U133A Array data from refine.bio (GSE83130) normalized by the SCAN method[36,37]. GBM RNA-seq upper quartile FPKM gene expression data came from the National Cancer Institute Genomic Data Commons TCGA Pan-Cancer Atlas[38,39]. Likewise, SNP and indel mutation calls came from the GDC Pan-Cancer Atlas MC3 public MAF (v0.2.8)[40]. We consider these data to be the products of standard processing pipelines for their respective DNA and RNA expression platforms. Our methods focus on cross-platform normalization and assume that any within-platform batch effects have already been corrected upstream.

For the purpose of these analyses, we restricted the data set to the tumor samples (termed 'matched samples') that were measured on both platforms (520 matched samples for BRCA, 150 for GBM). Mutation calls were missing for 12/520 BRCA samples and 4/150 GBM samples. A total of 16146 genes were measured on both expression platforms for BRCA and 11414 genes for GBM.

**Subtype and mutation status prediction and unsupervised feature extraction**
*Experimental design*. An overview of our experimental design for machine learning evaluations is illustrated in Fig. 1. Matched samples were split into training (2/3) and test (1/3) sets using the createDataPartition function in the caret package, which takes the balance of the class distributions in the training and holdout sets into account (i.e., stratified sampling; Fig. 1a)[41]. See Supplementary Fig. 1 for representative plots of subtype and mutation status distributions for BRCA and GBM. To create our panel of training sets, samples analyzed with RNA-seq were titrated into the training set via random selection in 10% increments to produce training sets containing 0%, 10%, 20%… 100% RNA-seq data (Fig. 1b). For every RNA-seq sample added to a training set, the matched microarray sample was removed, keeping the number and identity of patients consistent across all training data sets.

Machine learning methods are applied to each training set (there are 11 training sets, with 0–100% RNA-seq titrated in at 10% increments), which have each been normalized in various ways according to the details below (see Cross-platform normalization approaches and Creation of cross-platform training sets and single platform holdout sets). We then applied prediction models based on each training set to our holdout data sets. Two holdout sets were used: a set of RNA-seq data and a set of microarray data. We refer to these as the RNA-seq holdout set and microarray holdout set, respectively. These holdout sets (or test sets) of normalized microarray and RNA-seq data are kept separate and not titrated together in order to evaluate how the composition of the training set impacts prediction performance in each data type. The pipeline for partitioning data into training and testing, titration, and normalization was repeated 10 times using different random seeds, as were the downstream analyses (e.g., subtype and mutation status classification, unsupervised feature construction).

*Cross-platform normalization approaches*. In each randomized repeat of our experiment, the same set of patients was used across all training sets, with only the platform of the sample (either microarray or RNA-seq) varying across RNA-seq titration levels. Likewise, the microarray and RNA-seq holdout sets consisted of matched samples from the same set of patients. In accordance with how these methods would be used in practice, normalization was performed separately for training and holdout sets. For the normalization methods that require a reference distribution (e.g., QN and TDM), the RNA-seq training and holdout sets were normalized using the microarray portion of the training data as a reference, separately. This is analogous to the normalization process in the absence of matched samples; the training microarray data (which is 100, 90, … 0% of the training set, Fig. 1) would be used as a reference for both the RNA-seq training data and the RNA-seq holdout data. These normalization processes would be performed separately to avoid contamination of the holdout data and, thus, overestimation of performance. Only genes measured on both platforms were included, and any genes that had missing values in all samples or all equal values in the RNA-seq data (all samples in holdout data, training 'titration' samples at any sequencing level) were removed. See Supplementary Fig. 2 and its legend for an overview of how training and holdout normalization was performed for all methods.

Log₂-transformation (LOG): As the log₂-transformed array data contained negative values, the microarray data were inverse log-transformed and then log-transformed again after adding 1 to each expression value such that re-transformed values are non-negative. All missing values were set to zero. This array of data was used in all downstream processing steps.

Quantile normalization (QN): QN was performed using the preProcessCore R package[42]. Given the log₂-transformed microarray distribution (target), the normalize.quantiles.use.target method will normalize the columns (samples) of the RNA-seq data such that the data sets are drawn from the same distribution. For sets entirely composed of samples on a single platform without an applicable reference, normalize.quantiles was used.

Quantile normalization with CrossNorm (QN (CN)): CrossNorm[43] combines data from different distributions by stacking columns from paired samples or, in the more general case, combinatorially stacking columns from each distribution before performing quantile normalization. Before combining data sets, the array, and RNA-seq sample expression values were rescaled zero-to-one. The QN (CN) process differs from QN or QN-Z because RNA-seq samples are not normalized to fit the array distribution; rather, the samples from array and RNA-seq are cross-normalized toward the same shared distribution together. Since CrossNorm requires two data sets, there is no QN (CN) data for training at 0 and 100% RNA-seq. Likewise, QN (CN) array and RNA-seq test data is simply rescaled and quantile normalized without reference to array training data.

Quantile normalization followed by standardizing scores (QN-Z): This normalization method sequentially combines quantile normalization (QN) and z-score transformation (Z).

Training distribution matching (TDM): TDM was developed specifically to make RNA-seq test data compatible with models trained on microarray data[21]. It identifies the relationship that the microarray training data has between the spread of the middle half of the data and the extremes and then transforms the RNA-seq test set such that it has the same relationship between the spread of the middle half of the data and the extremes. Specifically, given a reference distribution (array data), TDM calculates the reference interquartile range (IQR), and then finds the number of times the reference IQR fits into the reference distribution's upper (max—Q3) and lower (Q1—min) quartiles. Then, TDM sets the new upper and lower bound of the target distribution (RNA-seq) by stretching or shrinking the target upper and lower quartiles by the IQR factors calculated from the reference distribution. Target distribution values are winsorized to match the new upper and lower bounds. The target distribution is then rescaled between 0 and 1, stretched to match the range of the reference distribution, and shifted to match the minimum value of the reference distribution.

We used the TDM R package to perform TDM normalization. For the training set composed of 0% RNA-seq data, models trained on log₂-transformed microarray data were used for prediction/reconstruction on the TDM normalized RNA-seq holdout set. TDM was not used when the training set consisted of 100% RNA-seq data, as there is no relevant microarray data to use as the reference distribution in this case.

Non-paranormal normalization (NPN): Our implementation of NPN is a rank-based inverse normal transformation that forces data to conform to a normal distribution by rank-transformation followed by quantile normalization, placing each observation where it would fall on a standard normal distribution[44]. NPN was performed using the huge R package prior to concatenating samples from both platforms and separately on single-platform holdout sets[45].

Standardizing scores (Z): z-scoring was performed on a per gene basis using the scale function in R prior to concatenating training samples from both platforms. Single-platform holdout sets were z-scored separately. Z-scores or standard scores are calculated $z = \frac{x-\mu}{\sigma}$, where $\mu$ and $\sigma$ are the gene mean and standard deviation, respectively.

Gene expression values were standardized to the range [0, 1] (zero-to-one transformation) on a per gene basis, either before or after concatenating samples from each platform.

As an additional basis for comparison, untransformed data (UN) was included in most analysis tasks. Untransformed array data is LOG data with no zero-to-one transformation. Untransformed RNA-seq data has no zero-to-one transformation.

Other approaches to combining and normalizing gene expression data are commonly used in single-cell RNA-sequencing studies to integrate data sets by reducing batch effects across studies and experimental conditions. We designed an approach analogous to Seurat's integration pipeline[46–50] in which single cells became single (bulk) samples, and our array and RNA-seq data sets were the two batches or experimental conditions. Due to low sample numbers at the edges of our titration protocol, many experimental conditions could not be integrated. For those conditions with enough samples from array and RNA-seq for successful integration, we have included machine learning prediction results in table form (see Data availability).

*Creation of cross-platform training sets and single platform holdout sets*. Training data sets are mixtures of microarray and RNA-seq samples titrated at 10% intervals, starting with 0% RNA-seq, then 10%, 20%, …, and 100%. When RNA-seq samples are added to a training set, the matched microarray samples are removed. At each titration level, microarray and RNA-seq samples are normalized according to various methods.

0% RNA-seq (100% microarray) training data: LOG data is the input to non-paranormal (NPN), quantile (QN), QN followed by Z (QN-Z), and z-score (Z) normalization. UN array data is LOG data without zero-to-one transformation. There is no QN (CN) or TDM for 100% array training data.

X% RNA-seq (100-X% microarray) training data: For LOG, microarray samples are LOG and RNA-seq values are $log_2(x + 1)$ transformed, then array and RNA-seq values undergo gene-level zero-to-one transformation separately before being joined together. For NPN, array and RNA-seq values undergo gene-level non-paranormal normalization separately, then get combined before gene-level zero-to-one transformation. For QN, RNA-seq data is transformed to align with the quantiles of array data, array and RNA-seq undergo gene-level zero-to-one transformation separately and are then combined. For QN (CN), each array and RNA-seq column (sample) is scaled zero-to-one and then normalized using the unpaired CrossNorm algorithm, followed by gene-level zero-to-one transformation. For QN-Z, array and RNA-seq data are QN and Z normalized separately before getting combined for gene-level zero-to-one transformation. Similarly, for TDM, we transform the spread of RNA-seq data to match the spread of array data, array and RNA-seq undergo gene-level zero-to-one transformation separately and are then combined. For Z, RNA-seq and array data are z-scored separately at the gene-level, then combined for gene-level zero-to-one transformation. For UN, RNA-seq data remains untransformed and data do not undergo gene-level zero-to-one transformation.

100% RNA-seq (0% microarray) training data: Untransformed RNA-seq data gets LOG, NPN, QN, QN-Z, and Z transformed by itself without reference to array data. It then undergoes gene-level zero-to-one transformation. For UN, RNA-seq data remains untransformed and data do not undergo gene-level zero-to-one transformation after each normalization. There is no QN (CN) or TDM applied to 100% RNA-seq training data.

Microarray test data (100% microarray): Like the 100% array training data, array test data gets LOG, NPN, QN, QN (CN), QN-Z, UN, and Z transformed by itself without reference to any other data, and it then undergoes gene-level zero-to-one transformation after each normalization except UN. There is no TDM for 100% array test data.

RNA-seq test data (100% RNA-seq): For LOG, NPN, QN (CN), and Z, RNA-seq values are transformed without reference to any other data. We transform the 100% RNA-seq test data using QN, QN-Z, and TDM across the RNA-seq titration level spectrum (0–90%) to match the training LOG array data associated with each titration level (0% through 90%), just as the training RNA-seq data was transformed to match the training LOG array data at each level. QN and QN-Z use no reference for 100% RNA-seq test data, and there is no TDM applied to 100% RNA-seq test data. Transformed data undergoes gene-level zero-to-one transformation after each normalization except UN.

*Subtype and mutation status prediction on mixed platform data sets.* Subtype classifications—derived from TCGA's analysis of the microarray data, rather than a clinical assay—were used as the subtype labels for supervised analyses (Fig. 1c)[29–31]. Because these labels were derived from the microarray data, analyses that aim to compare the subtype prediction quality of microarray- and sequencing-based platforms would be inappropriate. Our evaluations are restricted to the relative performance of normalization methods across both platforms.

We performed fivefold cross-validation on training sets for model training and hyperparameter optimization using total accuracy for performance evaluation. We trained the following three classifiers: LASSO logistic regression[51], linear support vector machine (SVM), and random forest. We used the glmnet R package implementation of LASSO[52]. The training of SVM and random forest classifiers was performed using the caret R package and utilizing the kernlab[53] and ranger[54] R packages, respectively. We used the Kappa statistic to evaluate performance on holdout data for two main reasons. The first reason is we make no assumptions about class balance in our data, so to mitigate the potential for bias due to class imbalance, we chose a metric that builds in a baseline probability of chance agreement. The second reason is to accommodate our multi-class outcome (e.g., five subtypes in both breast cancer and glioblastoma), which Kappa achieves without relying on a composite of one-vs-all comparisons, as classes were not balanced. Briefly, the Kappa statistic takes into account the expected accuracy of a random classifier and is generally considered to be less misleading than observed accuracy alone[55]. The formula for Cohen's Kappa is:

$$\kappa = \frac{p_0 - p_e}{1 - p_e} \qquad (1)$$

where $p_0$ is the agreement observed between two methods and $p_e$ is the expected probability of agreement by chance. In addition to the Kappa statistic, we used a composite one-vs-all approach to calculate several metrics designed for binary classification tasks with balanced data, including the area under the receiver operator curve (AUC), sensitivity, and specificity. Metrics for all subtype and mutation prediction models may be found in Supplementary Data 1.

*TP53* and *PIK3CA* mutation calls, including SNPs and indels, were derived from TCGA's MC3 public call set (v0.2.8)[40]. Our evaluation approach changed slightly for predicting mutation status. Given the higher prevalence of *TP53* and *PIK3CA* mutations in some expression-based subtypes, we wanted to make sure our mutation classifiers were not just predicting subtypes as a proxy for mutation status. We accounted for this by calculating the change in Kappa (delta Kappa) between a null model and a model built with true labels. Our null model was built with mutation labels randomized within each subtype. The null model Kappa represents the ability of the classifier to differentiate subtypes based on the prevalence of mutations in that subtype. Positive delta Kappa values indicate the ability of the model to predict mutation status beyond what would happen given the association of each mutation with particular subtypes.

*Unsupervised feature extraction from mixed platform data sets*
Pathway-level information extractor (PLIER): We used PLIER to identify coordinated patterns of expression in our data, which may be the result of gene regulation[24]. PLIER uses a set of biological constraints defined by gene expression pathways to decompose an expression matrix with non-negative matrix factorization. One output is a set of latent variables and loadings, and some of the input pathways may be associated with one or more latent variables, suggesting that the pathway plays a role in regulating gene expression. We ran PLIER using training data to assess the downstream benefit of creating larger data sets through cross-platform normalization. We report the proportion of significant pathways out of the global set of pathways, with higher proportions indicating a more complete capture of real biological variation. As a negative control, we identified the baseline false positive rate of significant results obtained by chance by permuting the gene-pathway relationships represented in the gene-pathway input matrix. Specifically, we permuted the gene (rows) and pathways (columns) matrix by resampling without replacement within each column such that the number of genes associated with each pathway remained constant. We compared results from single platform data sets (microarray or RNA-seq only) with normalized data comprising 50% microarray and 50% RNA-seq samples. For the single platform data sets, we included some with half of the available samples and some with the full set of samples. The normalized data sets comprised the full set of samples with half coming from either platform. The global set of biological pathways included 817 canonical, oncogenic, cell line, and blood cell pathways. For each input gene expression data set, we identified the total number of pathways significantly associated (FDR < 0.05) with at least one latent variable. To evaluate the potential benefit of increasing sample size to pathway stability and consistency, we identified oncogenic pathways that were more frequently (change in proportion > = 0.2) associated with at least one latent variable in the full sample size data sets compared to the half sample size data sets.

Principal components reconstruction for subtype prediction: We hypothesized that normalization methods may distort the utility of low-dimensional projections and that this distortion may differ by the level of RNA-seq data titrated into each training set. To test this, we evaluated machine learning tasks after projecting the test data onto the low-dimensional space defined by the training data. We performed principal components analysis (PCA) on each training set using the prcomp function in R, setting the number of components to 50 (Fig. 1c). We then projected the holdout data onto the training data PC space and reconstructed the holdout data using the first 50 principal components. We assessed reconstruction error (comparing holdout input, $y$, to reconstructed values, $\hat{y}$) by calculating the mean absolute scaled error (MASE) for each gene[56]. We selected mean absolute scaled error (MASE) to assess reconstruction error based on the needs of our application. Scale invariance is a key feature enabling comparison across multiple sequencing types and normalization methods. MASE is well suited for data sets with predicted values close to zero, in contrast with other measures. MASE also measures error in absolute terms, matching our preference to weight over and under estimation of values equally. MASE is calculated on a per gene basis as follows:

$$MASE = mean\left(\frac{|y_i - \hat{y}_i|}{\frac{1}{N}\sum_{i=1}^{N}|y_i - \bar{y}|}\right) \qquad (2)$$

where, for each sample $i$ (1 through N), $y_i$ is the original expression value, $\hat{y}_i$ is the expression value after reconstruction, and $\bar{y}$ is the average of the original expression values for that gene.

We performed supervised analysis following reconstruction to assess whether the subtype signals were retained or if features were dominated by noise introduced by combining platforms. We used the models trained for subtype classification to predict on the reconstructed holdout sets to assess how well the molecular subtype signal was retained in the reconstructed holdout data (Fig. 1c). We again used the Kappa statistic to evaluate performance.

**Statistics and reproducibility**. Statistical analyses were performed using R version 4.1.2 in a publicly available Docker image. Please refer to our source code (v2.3) on Github at https://www.github.com/greenelab/RNAseq_titration_results for instructions on how to access the Docker container, download data, and get started

running analyses, including a bash script to reproduce the entire analysis. We set seeds for reproducibility to account for stochastic elements in our code.

For each cancer type and predictor variable (subtype or mutation status), we created 10 replicate training and test data sets for machine learning and downstream analyses. Training and test data were kept separate from the start of each replicate. We used createDataPartition from R's caret package to balance the distribution of classes between training and test data. For a single replicate, two-thirds of available samples were allocated to training and one-third to test data. The exact number of samples partitioned to training and test data sets may vary slightly between replicates due to stratified sampling. For subtype prediction, 520 matched samples were available for breast cancer and 150 matched samples were available for glioblastoma (Supplementary Fig. 1). For mutation prediction (*PIK3CA* and *TP53*), 508 matched samples were available for breast cancer and 146 matched samples were available for glioblastoma.

To create data sets with varying levels of RNA-seq data, RNA-seq samples were randomly titrated into the array samples at 10% intervals to create 11 distinct training sets per replicate (0% through 100% RNA-seq). Importantly, as RNA-seq samples were titrated into the training data, the corresponding array sample was removed such that one patient's data was only represented by one sample per experimental repeat.

**Reporting summary**. Further information on research design is available in the Nature Portfolio Reporting Summary linked to this article.

## Data availability

All data that support the findings of this study are available on Figshare Plus at https://doi.org/10.25452/figshare.plus.19629864[57]. Data and code used to create data plots are available on Figshare at https://doi.org/10.6084/m9.figshare.19686453[58].

## Code availability

Source code (v2.3) is available on Github at https://www.github.com/greenelab/RNAseq_titration_results under a BSD-3 license, or https://doi.org/10.6084/m9.figshare.1970134[59]. The computational environment necessary to replicate our results are available in a Docker container. Instructions for how to access the Docker container, download data, and get started running analyses are available on our Github page.

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

## Acknowledgements
We thank Jake Crawford and Gregory Way for helpful code review and Amy Campbell for proofreading of the manuscript. This work was supported by the Gordon and Betty Moore Foundation [GBMF 4552], Alex's Lemonade Stand Foundation [GR-000002471], and the National Institutes of Health [T32-AR007442, U01-TR001263, R01-CA237170, K12GM081259, R01-HG010067].

## Author contributions
S.M.F. and J.N.T. wrote the manuscript and performed analyses. C.S.G. and J.N.T. conceptualized and oversaw the work.

## Competing interests
The authors declare the following competing interests: J.N.T. is a full time employee of Alex's Lemonade Stand Foundation, a funder of this work. The other authors declare no competing interests.
