## [Peer Review File · Communications Biology]

Reviewers' comments:

Reviewer #1 (Remarks to the Author):

Summary:

In this paper, the authors conduct a series of experiments to examine five normalization approaches (LOG, NPN, QN, TDM, and Z score) to combining microarray and RNA-seq data. To assess which normalizations are best suited for the data combined from different platforms, they perform supervised/unsupervised machine learning and differential expression analysis.

While I found interesting on the examination of the cross-platform normalization for machine learning and downstream differential expression analysis, I have some concerns about the technical method. In addition, using one data set to test the effectiveness of normalization is not very convincing to me, since the best-suited normalization approach often varies with data. However, I acknowledge that to find a universally accepted normalization approach is challenging. For readers who would like to use microarray and RNA-seq data jointly, I do think the paper conveys a clear message on which normalization method is more appropriate for different analysis tasks. If the authors could investigate the robustness and stability of different normalization approaches with more examples, the results will be more informative for readers.

Major comments:

1. For the prediction accuracy on holdout data in the classification task, why did the author use the Kappa statistic as the only assessment measure? Other evaluation metrics such as ROC curves, precision-recall curves, and area under the curve (AUC) would be more informative.
2. Again in the classification task, it is unclear whether the authors applied the classification methods to the microarray and RNA-seq data jointly or separately. I think the microarray and RNA-seq data should be used jointly to demonstrate how the normalization works, but it seems that the authors used the two data types separately.
3. It would be better to elaborate a bit more on why MASE is a good measure to assess reconstruction error.
4. "For the normalization experiments, we identified DEGs in data sets that were 50% microarray and 50% RNA-seq data", it would be interesting to have the performance of different normalization approaches if the data sets are not half-and-half balanced.
5. To quantify the similarity between the clusters generated from different normalization methods and the ground truth, Jaccard index is used to examine whether the clustering results are robust to the selection of samples, but it would be good to also report the Rand index between the clusters and the ground truth, which is also often used in many clustering studies to reflect the correctness of clustering.
6. Section 2.1. "...we expect that within-platform batch correction, if appropriate, should be performed prior to applying any of the evaluated cross-platform normalization methods". Is it valid to apply multiple layers of normalization to the data sets? I would prefer methods that jointly model these effects together.
7. Section 2.2.2. "For the normalization methods that require a reference distribution (e.g., QN and TDM), the RNA-seq training and holdout sets were normalized using the microarray data training data as a reference, separately." Why can microarray data be used as a reference for RNA-seq data? The authors may need to explain how the "reference distribution" is used.

8. Section 2.4.4. The authors should clearly define the notations in the formula.

9. The authors have used many methods/R packages and such information is scattered. It would be clearer to have a summary or figure showing the pipeline.

Minor comments:

Typo: page 2 line 10, "affecte" should be "affects"

Reviewer #2 (Remarks to the Author):

The paper presents an evaluation of different cross-platform normalisation methods for machine learning with mixed RNA-seq and microarray data based on classification of two breast cancer subtypes in TCGA data. No new methods are presented and the main novelty is in the experimental setup of changing the ratio of microarray vs. RNA-seq data in the training set.

While the evaluation and the conclusions reached are interesting, I feel the evaluation is still too limited to represent an advance in understanding which may influence thinking in the field, as required for publication in Communications Biology. The paper focuses on a single binary classification task between two quite distinct cancer types on a large and homogeneous data set. However, most applications where one would want to combine RNA-seq and microarray data are likely quite different from this: the questions can be more complex and the data sets more diverse. As a result, I do not feel the weight of the evidence provided in the paper is sufficient to support the broad claim made in the title.

The writing of the paper is mostly OK. A more mathematical presentation of the compared normalisation methods would help to really understand their operation.

Detailed comments:

1. I commend the authors for publishing their code. I downloaded it and tried to run it quickly, but was defeated by missing dependencies I did not have the time to resolve. You could make the package more user friendly by adding a script to easily install all dependencies.

2. The authors use a lot of space in the introduction for citing statistics on RNA-seq and microarray popularity that are 2 years old. For a paper of this level, surely you could spend an hour updating this with current statistics.

3. I find the \log_2 transformation proposed in Sec. 2.2.2 odd. If I understand correctly, there is no absolute unit for microarray data and therefore we could multiply the absolute numbers by an arbitrary constant, which would be equivalent to adding an arbitrary constant to the log scale values. The proposed transformation very strongly compresses small values which, depending on the range of values actually used, may or may not contain useful information.

4. Using both PCA and ICA for unsupervised comparison is pointless, because they will both return identical results (subject to random fluctuations). This is because ICA finds 50 components by first performing PCA to find 50 components and then only finds an orthogonal rotation among these. The reconstructions of the original data space from these components will be identical. In a sense it is reassuring the authors also confirmed this empirically, but I would highly recommend leaving out ICA because it is completely redundant.

5. In terms of differential expression analysis, the set of genes called will depend on two things:

ranking of the genes and the chosen cutoff. As it is difficult even in theory to match the cutoffs across platforms because of their different sensitivity characteristics, I would recommend additionally comparing differential expression results in terms of rankings provided by different methods to avoid potential confounding from differences in sensitivity.

6. In order to strengthen your paper for a future submission, I would highly recommend repeating the analysis with more diverse data and data sets of different sizes. Especially the latter could be achieved very easily by subsetting the current data.

Minor comments:

7. In Sec. 2.2.2 you write: "genes that had missing values in the RNA-seq data in all samples (...) were removed". This comment seems very strange, because as you state yourself in the introduction, RNA-seq always measures all genes. I would assume the difference was caused by mismatches in the annotation between RNA-seq and microarray, but I would suggest deleting the statement as it seems redundant given that you already say that only genes measured on both platforms were included.

Reviewer #3 (Remarks to the Author):

The authors compare different normalization approaches with respect to combining the two most-popular platforms for transcriptome profiling. Synergistic integration of datasets from the two platforms has good potential to improve our understanding of transcriptome dynamics, because each has its own strength and weakness. But it is not a trivial problem and there exist many options including the authors' original approach, TDM (proposed in Thompson et al 2016 PeerJ).

The manuscript may improve if the authors address the following.

[Major points]

(1) Find some evidence that combining data would benefit downstream analyses. In fact, Figure 5A may look a bit misleading on this matter because, as mixing proportion increases, the number of recovered silver standard DEGs decreases in many normalization methods. Consider, for example, comparing DEGs identified by microarray only, RNA-Seq only, and combined (50/50) dataset, and see if it improves enrichment of known signaling pathways.

(2) In order to combine unmatched samples from the two platforms, it may be important to find a nonlinear mapping function between microarray probe intensities and RNA-Seq read counts. A good mapping function (or transformation) would project data points from both platforms onto a new manifold such that the dynamic range is maximized. Since the authors already have samples that are measured on both platforms, I recommend visualizing data points in scatter plots (intensity vs read count in log scale) and see how it changes after each normalization. For example, please see Figure 2 of Zhao et al PLoS ONE 2014 doi:10.1371/journal.pone.0078644.

(3) Test and/or speculate on which normalization method would be suitable for eQTL analysis.

(4) If we transform read counts such that the distribution of normalized microarray intensities does not change, all the available microarray analysis pipelines should simply work fine. Isn't QN (and other well-performing normalizer) somewhat close to doing that?

[Minor points: to improve readability]

(1) Typos: p.4 "affecte" -> "affects", p.11 "that that" -> "that"

(2) Please describe in more detail what the rationale is behind creating "reconstructed" holdout datasets and then fitting classifiers in PCA/ICA tests. I found it was complex and a bit difficult to understand.

(3) Please describe what NPN does in more detail so that readers do not have to look up the reference to it. How is it different from Z or rank-based inverse normal transformations as in Beasley et al 2009 doi:10.1007/s10519-009-9281-0?

(4) Provide formula for kappa statistics.

(5) [Section 3.2] " We found that projecting the holdout sets onto QN and TDM normalized training space results in less loss of subtype classifier performance": Please rephrase using a quantitative metric.

(6) [Figure S1] Consider showing composition rather than counts if your goal is to show that the subtype distribution are close to each other.

(7) [Figure 3, S4, and S5] Put y-axis in a same scale on the same set of grids so readers can directly compare between subplots.

(8) Specify what reference is used for LOG normalizing microarray holdout set.

(9) When RNA-Seq samples are titrated, are the corresponding samples excluded from the microarray samples?

Reviewer #1 (Remarks to the Author):

Summary:

In this paper, the authors conduct a series of experiments to examine five normalization approaches (LOG, NPN, QN, TDM, and Z score) to combining microarray and RNA-seq data. To assess which normalizations are best suited for the data combined from different platforms, they perform supervised/unsupervised machine learning and differential expression analysis.

While I found interesting on the examination of the cross-platform normalization for machine learning and downstream differential expression analysis, I have some concerns about the technical method. In addition, using one data set to test the effectiveness of normalization is not very convincing to me, since the best-suited normalization approach often varies with data. However, I acknowledge that to find a universally accepted normalization approach is challenging. For readers who would like to use microarray and RNA-seq data jointly, I do think the paper conveys a clear message on which normalization method is more appropriate for different analysis tasks. If the authors could investigate the robustness and stability of different normalization approaches with more examples, the results will be more informative for readers.

We appreciate the reviewer's contribution to improving the analyses presented in our manuscript. We have acted on the reviewer's advice to examine the effectiveness of cross-platform normalization in a broader range of applications as well as added clarity to many of our methods descriptions.

Namely, we have expanded the cancer cohorts being analyzed to include both breast cancer (BRCA) and glioblastoma (GBM) from The Cancer Genome Atlas (TCGA). Additionally, we have added additional prediction tasks -- now, we examine subtype prediction as well as TP53 and PIK3CA mutation status prediction from expression data. We find consistent patterns across cancer types in the utility of cross-platform normalization in subtype prediction, and we examine the variability of the predictive value of TP53 and PIK3CA mutation status models.

We incorporated the reviewer's suggestion to explore the small sample size differentially expressed gene experiment across a range of titration levels and present those results below. Further, we added another unsupervised downstream analysis (PLIER) that identifies active pathways within the gene expression data, and we show the benefit of increasing sample size through cross-platform normalization. These major changes are reflected in significant updates to the main text and figures, as well as specific changes detailed below.

Major comments:

1. For the prediction accuracy on holdout data in the classification task, why did the author use the Kappa statistic as the only assessment measure? Other evaluation metrics such as ROC curves, precision-recall curves, and area under the curve (AUC) would be more informative.

We thank the reviewer for directing our attention to these alternative choices for measuring prediction performance. We chose the kappa statistic for two main reasons. The first reason is we make no assumptions about class balance in our data, so to mitigate the potential for bias due to class imbalance, we chose a metric that builds in a baseline probability of chance agreement. The second reason is to accommodate our multi-class outcome (e.g. five subtypes in both breast cancer and glioblastoma), which kappa achieves without relying on a composite of one-vs-all comparisons. We have added this justification to our methods discussion (lines 243-254):

We used the Kappa statistic to evaluate performance on holdout data for two main reasons. The first reason is we make no assumptions about class balance in our data, so to mitigate the potential for bias due to class imbalance, we chose a metric that builds in a baseline probability of chance agreement. The second reason is to accommodate our multi-class outcome (e.g., five subtypes in both breast cancer and glioblastoma), which Kappa achieves without relying on a composite of one-vs-all comparisons, as classes were not balanced. Briefly, the Kappa statistic takes into account the expected accuracy of a random classifier and is generally considered to be less misleading than observed accuracy alone.⁴⁵ The formula for Cohen's Kappa is:

$$\kappa = \frac{p_0 - p_e}{1 - p_e}$$

where p_0 is the agreement observed between two methods and p_e is the expected probability of agreement by chance.

2. Again in the classification task, it is unclear whether the authors applied the classification methods to the microarray and RNA-seq data jointly or separately. I think the microarray and RNA-seq data should be used jointly to demonstrate how the normalization works, but it seems that the authors used the two data types separately.

We appreciate the reviewer pointing out the need to clarify our experimental design and methods of normalization and application to test data. Training models were trained using datasets comprising both array and RNA-seq samples. Prediction models were evaluated using test data from single platforms. We have included a new supplementary figure (Figure S2) that illustrates how microarray and RNA-seq data are processed (see following page). We have also added additional details to the methods section to more fully communicate our process (lines 134-150):

2.2.2 Cross-platform normalization approaches

In each randomized repeat of our experiment, the same set of patients was used across all training sets, with only the platform of the sample (either microarray or RNA-seq) varying across RNA-seq titration levels. Likewise, the microarray and RNA-seq holdout sets consisted of matched samples from the same set of patients. In accordance with how these methods would be used in practice, normalization was performed separately for training and holdout sets. For the normalization methods that require a reference distribution (e.g., QN and TDM), the RNA-seq training and holdout sets were normalized using the microarray portion of the training data as a reference, separately. This is analogous to the normalization process in the absence of matched samples; the training microarray data (which is 100%, 90%, ... 0% of the training set, Fig 1) would be used as a reference for both the RNA-seq training data and the RNA-seq holdout data. These normalization processes would be performed separately to avoid “contamination” of the holdout data and, thus, overestimation of performance. Only genes measured on both platforms were included, and any genes that had missing values in all samples or all equal values in the RNA-seq data (all samples in holdout data, training ‘titration’ samples at any sequencing level) were removed. See Fig S2 and its legend for an overview of how training and holdout normalization was performed for all methods.

Supplementary Figure 2. Overview of cross-platform normalization with and without reference. Double hatched boxes refer to data processed using microarray data as a reference distribution. Single hatched boxes refer to data processed without using microarray data as a reference (no reference). Orange boxes refer to microarray data. Blue boxes refer to RNA-seq data. (A) Matched samples are split into a training set and testing set. (B) Each training set is composed of between 0% and 100% RNA-seq data. For No Reference, the microarray portion and RNA-seq portion are normalized separately for LOG, NPN, and Z. With Reference, the microarray portion remains unprocessed (LOG) while the RNA-seq portion is normalized using the microarray portion as a reference distribution for QN, QN-Z, and TDM. Microarray and RNA-seq portions are then combined to form each cross-platform normalized training data set. (C) Microarray test data normalization occurs without reference for LOG, NPN, QN, QN-Z, and Z, and with-out reference for RNA-seq data for LOG, NPN, and Z. RNA-seq test data normalization uses microarray training data as the reference distribution for QN, QN-Z, and TDM at each percentage of RNA-seq titration in training. (LOG - log₂-transformed; NPN - nonparanormal normalization; QN - quantile normalization; QN-Z - quantile normalization followed by z-score; TDM - Training Distribution Matching; UN - untransformed; Z - z-score)

3. It would be better to elaborate a bit more on why MASE is a good measure to assess reconstruction error.

We selected mean absolute scaled error (MASE) to assess reconstruction error based on the needs of our application. Scale invariance is a key feature enabling comparison across multiple sequencing types and normalization methods. MASE is well suited for data sets with predicted values close to zero, in contrast to other measures. MASE also measures error in absolute terms, matching our preference to count over and under estimation the same. We have added this discussion to the definition of MASE in the methods section (lines 288-298):

We assessed reconstruction error (comparing holdout input, y , to reconstructed values, \hat{y}) by calculating the mean absolute scaled error (MASE) for each gene. We selected mean absolute scaled error (MASE) to assess reconstruction error based on the needs of our application. Scale invariance is a key feature enabling comparison across multiple sequencing types and normalization methods. MASE is well suited for data sets with predicted values close to zero, in contrast with other measures. MASE also measures error in absolute terms, matching our preference to weight over and under estimation of values equally. MASE is calculated on a per gene basis as follows:

$$MASE = \text{mean} \left(\frac{|y_i - \hat{y}_i|}{\frac{1}{N} \sum_{i=1}^N |y_i - \bar{y}|} \right)$$

where, for each sample i (1 through N), y_i is the original expression value, \hat{y}_i is the expression value after reconstruction, and \bar{y} is the average of the original expression values for that gene.

4. “For the normalization experiments, we identified DEGs in data sets that were 50% microarray and 50% RNA-seq data”, it would be interesting to have the performance of different normalization approaches if the data sets are not half-and-half balanced.

We appreciate the reviewer's idea to evaluate performance across the titration percentage spectrum. We modified our approach to incorporate this advice and present the results below. We find there is little difference in results from datasets with 30%, 50%, and 70% RNA-seq data, rather it is having enough sample size that impacts results. (We also checked at 10-90% RNA-seq and saw no difference.) Given the addition of GBM data to our project, we also analyzed Classical vs. Mesenchymal subtypes within GBM in addition to Her2 vs. LumA in BRCA. We have updated this figure in the manuscript (Figure 6B, Figure S13) and more fully describe how our small n datasets were constructed (lines 325-332):

2.3.2 Small sample size experiment

For each number of samples (n), we randomly selected n ($n = 3, 4, 5, 6, 8, 15, 25, 50$) samples each from two BRCA subtypes (Her2 and LumA) and up to 25 samples from GBM subtypes Classical and Mesenchymal, due to lower sample availability in GBM. To generate platform-specific silver standards, we compared the $2n$ samples in single platform data. For the normalization experiments, we identified DEGs in data sets with varying levels of RNA-seq data titrated in (30%, 50%, and 70%). Ten repeats were performed. All genes with an FDR < 10% were considered differentially expressed.

For additional context, we required at least 3 array and 3 RNA-seq samples to create our datasets, so some levels of n did not meet that threshold. We selected n samples from each cancer subtype, then randomized those samples into which would come from array and which from RNA-seq, based on the % RNA-seq. Thus, for $n = 3$ (3 from each subtype, for a total of 6), at 30% RNA-seq, there were only 2 ($6 \cdot 0.3 = 1.8$, then rounded up to 2) RNA-seq samples available.

Figure 6. Overlap between platform-specific silver standard differentially expressed genes (DEGs) and experimental DEGs when testing (A) all samples and (B) smaller sample sizes. Genes differentially expressed between BRCA subtypes Her2 and LumA samples were identified. (A) All Her2 and LumA samples were compared for all normalization methods with varying amounts of RNA-seq data included. All genes with an FDR < 5% were considered differentially expressed. These lists were used to calculate the Jaccard index and Spearman rank correlation. (B) Small n experiment. For each number of samples (n), we randomly selected n samples each from the Her2 and LumA subtypes of BRCA. To generate platform-specific standards, we compared the 2n samples in single platform data (100% log₂-transformed microarray data and 100% RNA-seq data). For the normalization experiments, we identified DEGs in data sets composed of varying percentages of RNA-seq data (30%, 50%, and 70%). Ten repeats were performed. All genes with an FDR < 10% were considered differentially expressed. These lists were used to calculate the Jaccard index and Spearman rank correlation. The median and “95% confidence on the median” (+/-1.58 IQR/sqrt(n)) of each statistic from ten repeats is displayed.⁵² (LOG – log₂-transformed; NPN – nonparanormal normalization; QN – quantile normalization; QN-Z – quantile normalization followed by z-score; TDM – Training Distribution Matching; UN – untransformed; Z – z-score)

5. To quantify the similarity between the clusters generated from different normalization methods and the ground truth, Jaccard index is used to examine whether the clustering results are robust to the selection of samples, but it would be good to also report the Rand index between the clusters and the ground truth, which is also often used in many clustering studies to reflect the correctness of clustering.

We have carefully considered the reviewer's suggestion to include the Rand index in addition to the Jaccard similarity, and we thank them for this idea. In our specific case, the set of true negative genes (genes not found to be differentially expressed between two subtypes) could be very large. Thus, theoretically, the Rand index, which includes TN in both the numerator and the denominator, could be driven by this TN term when there are very few true positives, leading to a convergence of index values which would be hard to distinguish and interpret.

In testing motivated by the reviewer's comment, we found this to be true empirically in our experiment using small sample sizes. In contrast, the Jaccard index prizes positive overlap, and we found that to be a better measure of similarity in our case with such a large set of TN genes. For analysis on the full data sets, the difference in interpretation between Jaccard and Rand (and Spearman rank correlation) was negligible. Please see results below – Rand index scores are close to one when the set of true negatives determined by the silver standard data is high. Given the similarity in interpretation between Jaccard and Rand for large n data and the poor performance of Rand at small n, we chose to focus manuscript results on Jaccard scores and Spearman rank correlation.

BRCA Her2 vs. LumA FDR < 5%

Silver Standard Comparison Platform — Microarray — RNA-seq

Small n Experiment: BRCA Her2 vs. LumA FDR < 10%

6. Section 2.1. "...we expect that within-platform batch correction, if appropriate, should be performed prior to applying any of the evaluated cross-platform normalization methods". Is it valid to apply multiple layers of normalization to the data sets? I would prefer methods that jointly model these effects together.

We thank the reviewer for their careful reading and highlighting this confusing phrasing. In this context, our intention is that within-platform batch effects be corrected before cross-platform normalization. The steps required for batch-effect correction will be different across different experiments, so we have focused our analysis on cross-platform normalization after this batch-corrected common starting point has been reached. We have edited the language of the manuscript to make this distinction more clear (lines 101-103):

Our methods focus on cross-platform normalization and assume that any within-platform batch effects have already been corrected upstream.

7. Section 2.2.2. "For the normalization methods that require a reference distribution (e.g., QN and TDM), the RNA-seq training and holdout sets were normalized using the microarray data training data as a reference, separately." Why can microarray data be used as a reference for RNA-seq data? The authors may need to explain how the "reference distribution" is used.

We thank the reviewer for this opportunity to more clearly state how the reference distribution is used. The goal of quantile normalization (QN) and training distribution matching (TDM) is to map the target values (RNA-seq) onto the reference space (array). For QN, RNA-seq data is transformed to align with the quantiles of array data. For TDM, the spread of RNA-seq data is matched to the spread of array data. We have significantly expanded the methods section to clearly explain the utility of the reference distribution in QN and TDM in reshaping the target distribution. Please our new Figure S2 (above) and updated manuscript lines 155-179:

Quantile normalization (QN): QN was performed using the preProcessCore R package. Given the log₂-transformed microarray distribution ("target"), the normalize.quantiles.use.target method will normalize the columns (samples) of the RNA-seq data such that the data sets are drawn from the same distribution. For sets entirely composed of samples on a single platform without an applicable reference, normalize.quantiles was used.

Quantile normalization followed by standardizing scores (QN-Z): This normalization method sequentially combines quantile normalization (QN) and z-score transformation (Z).

Training Distribution Matching (TDM): TDM was developed specifically to make RNA-seq test data compatible with models trained on microarray data. It identifies the relationship that the microarray training data has between the

spread of the middle half of the data and the extremes and then transforms the RNA-seq test set such that it has the same relationship between the spread of the middle half of the data and the extremes. Specifically, given a reference distribution (array data), TDM calculates the reference interquartile range (IQR), and then finds the number of times the reference IQR fits into the reference distribution's upper (max - Q3) and lower (Q1 - min) quartiles. Then, TDM sets the new upper and lower bound of the target distribution (RNA-seq) by stretching or shrinking the target upper and lower quartiles by the IQR factors calculated from the reference distribution. Target distribution values are winsorized to match the new upper and lower bounds. The target distribution is then rescaled between 0 and 1, stretched to match the range of the reference distribution, and shifted to match the minimum value of the reference distribution.

We used the TDM R package to perform TDM normalization. For the training set composed of 0% RNA-seq data, models trained on log2-transformed microarray data were used for prediction/reconstruction on the TDM normalized RNA-seq holdout set. TDM was not used when the training set consisted of 100% RNA-seq data, as there is no relevant microarray data to use as the reference distribution in this case.

8. Section 2.4.4. The authors should clearly define the notations in the formula.

We have added notation to fully define the mean absolute scaled error (MASE) formula. In particular, we have defined the meaning of the index, i , and the observation size, N . We have also more clearly defined these values in terms of our specific application, namely that on a per gene basis, i is an index over the number of samples and N is the total sample size. Please see our updated text (lines 288-298):

We assessed reconstruction error (comparing holdout input, y , to reconstructed values, \hat{y}) by calculating the mean absolute scaled error (MASE) for each gene. We selected mean absolute scaled error (MASE) to assess reconstruction error based on the needs of our application. Scale invariance is a key feature enabling comparison across multiple sequencing types and normalization methods. MASE is well suited for data sets with predicted values close to zero, in contrast with other measures. MASE also measures error in absolute terms, matching our preference to weight over and under estimation of values equally. MASE is calculated on a per gene basis as follows:

$$MASE = \text{mean} \left(\frac{|y_i - \hat{y}_i|}{\frac{1}{N} \sum_{i=1}^N |y_i - \bar{y}|} \right)$$

where, for each sample i (1 through N), y_i is the original expression value, \hat{y}_i is the expression value after reconstruction, and \bar{y} is the average of the original expression values for that gene.

9. The authors have used many methods/R packages and such information is scattered. It would be clearer to have a summary or figure showing the pipeline.

We greatly appreciate the reviewer's encouragement for comprehensive and organized documentation of computational methods. We have created a supplementary figure to illustrate the workflow, including the more technical aspects (see new Figure S2 above). We have also developed a Docker image to enable replication of the analysis environment and reproduce the entire pipeline, from data download to figure output. We have updated the manuscript to include the new technical summary figure and also reference our updated public github repository with full details.

Minor comments:

Typo: page 2 line 10, "affecte" should be "affects"

We have updated the manuscript to fix this error. Thank you.

Reviewer #2 (Remarks to the Author):

The paper presents an evaluation of different cross-platform normalisation methods for machine learning with mixed RNA-seq and microarray data based on classification of two breast cancer subtypes in TCGA data. No new methods are presented and the main novelty is in the experimental setup of changing the ratio of microarray vs. RNA-seq data in the training set.

While the evaluation and the conclusions reached are interesting, I feel the evaluation is still too limited to represent an advance in understanding which may influence thinking in the field, as required for publication in Communications Biology. The paper focuses on a single binary classification task between two quite distinct cancer types on a large and homogeneous data set. However, most applications where one would want to combine RNA-seq and microarray data are likely quite different from this: the questions can be more complex and the data sets more diverse. As a result, I do not feel the weight of the evidence provided in the paper is sufficient to support the broad claim made in the title.

The writing of the paper is mostly OK. A more mathematical presentation of the compared normalisation methods would help to really understand their operation.

We thank the reviewer for their insights and perspective in shaping this manuscript. We have integrated their advice by expanding the data sets used to illustrate the methods of cross-platform normalization to include both breast cancer (BRCA - 520 patients) and glioblastoma (GBM - 150 patients) from The Cancer Genome Atlas (TCGA). Further, we have added new prediction tasks to explore the utility of cross-platform normalization. Now, in addition to the five-subtype classification of BRCA data, we add subtype classification in GBM (five subtypes) and TP53 and PIK3CA mutation status prediction in both cancer types. These major changes are reflected in significant updates to the main text and figures, as well as changes detailed below.

At the reviewer's suggestion, we have improved user-friendliness by containerizing our project code in a docker container and providing more guidance on our github repository. We have updated the current statistics about the availability of array and RNA-seq data and removed the redundant ICA analysis. The reviewer's insight about differentially expressed gene FDR cutoffs led us to include Spearman's rank correlation in that analysis to great benefit, because it showed that there was underlying similarity between DEG sets, even at small sample sizes, that was masked by the cutoff-dependent Jaccard metric.

We have also included a deeper technical explanation of the normalization methods presented, especially Training Distribution Matching (TDM) and nonparanormal normalization (NPN) as well as a more thorough visualization of the normalization approach (please see new Figure S2 below) (lines 162-183):

Training Distribution Matching (TDM): TDM was developed specifically to make RNA-seq test data compatible with models trained on microarray data. It identifies the relationship that the microarray training data has between the spread of the middle half of the data and the extremes and then transforms the RNA-seq test set such that it has the same relationship between the spread of the middle half of the data and the extremes. Specifically, given a reference distribution (array data), TDM calculates the reference interquartile range (IQR), and then finds the number of times the reference IQR fits into the reference distribution's upper (max - Q3) and lower (Q1 - min) quartiles. Then, TDM sets the new upper and lower bound of the target distribution (RNA-seq) by stretching or shrinking the target upper and lower quartiles by the IQR factors calculated from the reference distribution. Target distribution values are winsorized to match the new upper and lower bounds. The target distribution is then rescaled between 0 and 1, stretched to match the range of the reference distribution, and shifted to match the minimum value of the reference distribution.

We used the TDM R package to perform TDM normalization. For the training set composed of 0% RNA-seq data, models trained on log₂-transformed microarray data were used for prediction/reconstruction on the TDM normalized RNA-seq holdout set. TDM was not used when the training set consisted of 100% RNA-seq data, as there is no relevant microarray data to use as the reference distribution in this case.

Non-paranormal normalization (NPN): NPN forces data to conform to a normal distribution by rank-transformation followed by quantile normalization, placing each observation where it would fall on a normal distribution. NPN was performed using the huge R package prior to concatenating samples from both platforms and separately on single-platform holdout sets.

Supplementary Figure 2. Overview of cross-platform normalization with and without reference. Double hatched boxes refer to data processed using microarray data as a reference distribution. Single hatched boxes refer to data processed without using microarray data as a reference (no reference). Orange boxes refer to microarray data. Blue boxes refer to RNA-seq data. (A) Matched samples are split into a training set and testing set. (B) Each training set is composed of between 0% and 100% RNA-seq data. For No Reference, the microarray portion and RNA-seq portion are normalized separately for LOG, NPN, and Z. With Reference, the microarray portion remains unprocessed (LOG) while the RNA-seq portion is normalized using the microarray portion as a reference distribution for QN, QN-Z, and TDM. Microarray and RNA-seq portions are then combined to form each cross-platform normalized training data set. (C) Microarray test data normalization occurs without reference for LOG, NPN, QN, QN-Z, and Z, and without reference for RNA-seq data for LOG, NPN, and Z. RNA-seq test data normalization uses microarray training data as the reference distribution for QN, QN-Z, and TDM at each percentage of RNA-seq titration in training. (LOG - log₂-transformed; NPN - nonparanormal normalization; QN - quantile normalization; QN-Z - quantile normalization followed by z-score; TDM - Training Distribution Matching; UN - untransformed; Z - z-score)

Detailed comments:

1. I commend the authors for publishing their code. I downloaded it and tried to run it quickly, but was defeated by missing dependencies I did not have the time to resolve. You could make the package more user friendly by adding a script to easily install all dependencies.

We appreciate the reviewer's effort to run our code and have taken deliberate steps to improve the usability and adaptability of our analysis. We created a Docker image to handle all dependency installation and now give thorough usage instructions on our github page. Further, we added command line options to all scripts to enable users to work with other data in their own context. Our github repository has been adapted with new users in mind, including how to get started with our Docker image and run the pipeline.

2. The authors use a lot of space in the introduction for citing statistics on RNA-seq and microarray popularity that are 2 years old. For a paper of this level, surely you could spend an hour updating this with current statistics.

We thank the reviewer for this comment. We have expanded this section to include up-to-date statistics. After searching on GEO and ArrayExpress, we found the current ratio of summarized human array to RNA-seq data is close to 1:1 in 2022 and have updated the manuscript to reflect this (lines 32-36):

RNA-sequencing (RNA-seq) assays represent a growing share of new gene expression experiments. In February 2022, the ratio of summarized human microarray to RNA-seq samples from GEO and ArrayExpress was close to one (1.13:1), meaning that half of all summarized human samples come from either platform, although RNA-seq overtook microarray data as the leading source of new submissions to ArrayExpress in 2018.

3. I find the log₂ transformation proposed in Sec. 2.2.2 odd. If I understand correctly, there is no absolute unit for microarray data and therefore we could multiply the absolute numbers by an arbitrary constant, which would be equivalent to adding an arbitrary constant to the log scale values. The proposed transformation very strongly compresses small values which, depending on the range of values actually used, may or may not contain useful information.

We appreciate the reviewer's concern and have rewritten this section to more clearly communicate our meaning. Standard microarray technologies report expression values on the log₂ scale. Our purpose is to emphasize that we are working with log₂ scaled data. We first check to see if any values are negative. If so, we perform an inverse log transformation and then retransform on the log₂ scale with log₂(x+1) so the minimum value is 0. We then convert any NA values to 0 and rescale all expression values to fall between [0, 1]. Our new supplementary figure Figure S2 (see above) shows how the

log2 array data is used as the basis for much of the downstream normalization process. Please see our updated description in the manuscript (lines 151-154):

Log2-transformation (LOG): As the log2-transformed array data contained negative values, the microarray data was inverse log-transformed and then log-transformed such that values are non-negative by adding 1 to each expression value before transformation. All missing values were set to zero. This array data was used in all downstream processing steps.

4. Using both PCA and ICA for unsupervised comparison is pointless, because they will both return identical results (subject to random fluctuations). This is because ICA finds 50 components by first performing PCA to find 50 components and then only finds an orthogonal rotation among these. The reconstructions of the original data space from these components will be identical. In a sense it is reassuring the authors also confirmed this empirically, but I would highly recommend leaving out ICA because it is completely redundant.

We appreciate the reviewer's insight here, and we have removed independent component analysis (ICA) from our analysis, while keeping principal component analysis (PCA) in the low-dimensional reconstruction section of the manuscript. We agree that in our specific application, PCA and ICA offer the same information, and ICA has been removed as redundant.

5. In terms of differential expression analysis, the set of genes called will depend on two things: ranking of the genes and the chosen cutoff. As it is difficult even in theory to match the cutoffs across platforms because of their different sensitivity characteristics, I would recommend additionally comparing differential expression results in terms of rankings provided by different methods to avoid potential confounding from differences in sensitivity.

We appreciate the reviewer's concern for confounding due to sensitivity differences. We added Spearman rank correlation as an additional metric to our differentially expressed gene analyses, both with the full data and the small n experiment, and have included Jaccard and Spearman in all display figures. For analyses utilizing the full data sets, there was little difference in interpretation between Jaccard and Spearman (both showed similar trends across the % RNA-seq spectrum). However, for the small n experiment, we often returned very low Jaccard scores because no genes met the significance threshold while Spearman rank correlation showed there was underlying agreement between the silver standard and experimental data obscured by Jaccard. Please see Figure 6 below as an example of where we have included Spearman rank correlation as a metric of DEG similarity between silver standard and experimental results. Here is our updated method description of how the DEG experiments were conducted (lines 304-332):

2.3 Differential expression analyses

We used a standard two-group single-channel experimental design in *limma* to identify differentially expressed genes. We used Benjamini-Hochberg correction multiple hypotheses testing, the output of which is known as a false discovery rate (FDR). We used mean-variance modeling at the observational level (VOOM) as implemented in *limma* to pre-process the single platform RNA-seq data. We termed the set of differentially expressed genes (DEGs) detected at a specified FDR from single platform data microarray and RNA-seq "silver standards". As with our supervised and unsupervised learning tasks, this experiment was designed to evaluate differential expression analysis in the absence of matched samples. We titrated in RNA-seq samples at 10% increments and identified DEGs (Fig 2). We compared them by calculating the Jaccard similarity, J , between the silver standard DEGs, S , and the experimental DEGs, E , as follows:

$$J(S, E) = \frac{|S \cap E|}{|S \cup E|}$$

We also compared the silver standard and experimental results using rank-based correlation (Spearman's rho).

2.3.1 Large sample size experiment

For the large sample size experiment, we used all microarray data and all RNA-seq data to identify the silver standard DEGs. In BRCA, we defined two separate comparisons to identify DEGs between Basal subtype and all other subtypes, then repeated to identify DEGs between Her2 and LumA subtypes. In GBM, the comparisons were Proneural versus others and Classical versus Mesenchymal.

2.3.2 Small sample size experiment

For each number of samples (n), we randomly selected n ($n = 3, 4, 5, 6, 8, 15, 25, 50$) samples each from two BRCA subtypes (Her2 and LumA) and up to 25 samples from GBM subtypes Classical and Mesenchymal, due to lower sample availability in GBM. To generate platform-specific silver standards, we compared the $2n$ samples in single platform data. For the normalization experiments, we identified DEGs in data sets with varying levels of RNA-seq data titrated in (30%, 50%, and 70%). Ten repeats were performed. All genes with an FDR < 10% were considered differentially expressed.

Figure 6. Overlap between platform-specific silver standard differentially expressed genes (DEGs) and experimental DEGs when testing (A) all samples and (B) smaller sample sizes. Genes differentially expressed between BRCA subtypes Her2 and LumA samples were identified. (A) All Her2 and LumA samples were compared for all normalization methods with varying amounts of RNA-seq data included. All genes with an FDR < 5% were considered differentially expressed. These lists were used to calculate the Jaccard index and Spearman rank correlation. (B) Small n experiment. For each number of samples (n), we randomly selected n samples each from the Her2 and LumA subtypes of BRCA. To generate platform-specific standards, we compared the 2n samples in single platform data (100% log₂-transformed microarray data and 100% RNA-seq data). For the normalization experiments, we identified DEGs in data sets composed of varying percentages of RNA-seq data (30%, 50%, and 70%). Ten repeats were performed. All genes with an FDR < 10% were considered differentially expressed. These lists were used to calculate the Jaccard index and Spearman rank correlation. The median and “95% confidence on the median” (+/-1.58 IQR/sqrt(n)) of each statistic from ten repeats is displayed.⁵² (LOG – log₂-transformed; NPN – nonparanormal normalization; QN – quantile normalization; QN-Z – quantile normalization followed by z-score; TDM – Training Distribution Matching; UN – untransformed; Z – z-score)

6. In order to strengthen your paper for a future submission, I would highly recommend repeating the analysis with more diverse data and data sets of different sizes. Especially the latter could be achieved very easily by subsetting the current data.

We thank the reviewer for suggesting this new direction. We have added an additional cancer type – glioblastoma (GBM) from The Cancer Genome Atlas – to our analysis. GBM is a smaller data set than our BRCA data (150 samples in GBM vs. 520 in BRCA). Adding GBM brings biological diversity to our analysis because GBM subtypes are driven by different mechanisms than BRCA subtypes. Please see our updated data set description in the manuscript (lines 87-108):

2.1 Evaluation gene expression and mutation data

The Cancer Genome Atlas (TCGA) breast cancer (BRCA) and glioblastoma (GBM) data sets include samples that have been measured with both microarray and RNA-seq platforms. In addition, both BRCA and GBM have well-defined molecular subtypes that are suitable for use as labels/classes for supervised machine learning approaches we describe below. BRCA and GBM also show recurrent TP53 and PIK3CA gene mutations, allowing for the prediction of mutation status based on gene expression profiles. For BRCA (520 pairs of matched samples), we used log2-transformed, lowess normalized Agilent 244K microarray data and RSEM (RNA-seq by Expectation Maximization) gene-level count RNA-seq data. For GBM (150 pairs of matched samples) we obtained Affymetrix HT Human Genome U133A Array data from refine.bio (GSE83130) normalized by the SCAN method. GBM RNA-seq upper quartile FPKM gene expression data came from the National Cancer Institute Genomic Data Commons TCGA Pan-Cancer Atlas. Likewise, SNP and indel mutation calls came from the GDC Pan-Cancer Atlas MC3 public MAF (v0.2.8). We consider these data to be the products of standard processing pipelines for their respective DNA and RNA expression platforms. Our methods focus on cross-platform normalization and assume that any within-platform batch effects have already been corrected upstream.

For the purpose of these analyses, we restricted the data set to the tumor samples (termed ‘matched samples’) that were measured on both platforms (520 matched samples for BRCA, 150 for GBM). Mutation calls were missing for 12/520 BRCA samples and 4/150 GBM samples. A total of 16146 genes were measured on both expression platforms for BRCA and 11414 genes for GBM.

Minor comments:

7. In Sec. 2.2.2 you write: "genes that had missing values in the RNA-seq data in all samples (...) were removed". This comment seems very strange, because as you state yourself in the introduction, RNA-seq always measures all genes. I would assume the difference was caused by mismatches in the annotation between RNA-seq and microarray, but I would suggest deleting

the statement as it seems redundant given that you already say that only genes measured on both platforms were included.

We appreciate the reviewer's close reading, which led us to modify language used to describe our data processing. We have updated the text to read as follows (lines 146-149):

Only genes measured on both platforms were included, and any genes that had missing values in all samples or all equal values in the RNA-seq data (all samples in holdout data, training 'titration' samples at any sequencing level) were removed.

Reviewer #3 (Remarks to the Author):

The authors compare different normalization approaches with respect to combining the two most-popular platforms for transcriptome profiling. Synergistic integration of datasets from the two platforms has good potential to improve our understanding of transcriptome dynamics, because each has its own strength and weakness. But it is not a trivial problem and there exist many options including the authors' original approach, TDM (proposed in Thompson et al 2016 PeerJ).

The manuscript may improve if the authors address the following.

We thank the reviewer for their effort dedicated to improving this work. We second their optimism that the integration of array and RNA-seq gene expression data can lead to valuable research gains. We have included a wide range of normalization techniques to illustrate the variety of potential methods in this approach. We have also included additional cancer types (now including glioblastoma [GBM] from The Cancer Genome Atlas) and prediction tasks (TP53 and PIK3CA mutation status prediction) to broaden applicability.

Importantly, we have shown more evidence that increasing sample size through cross-platform normalization improves downstream analyses in a new PLIER pathway analysis. We have also improved the manuscript based on the reviewer's comments regarding formula, visualization, and methods details. Thank you!

[Major points]

(1) Find some evidence that combining data would benefit downstream analyses. In fact, Figure 5A may look a bit misleading on this matter because, as mixing proportion increases, the number of recovered silver standard DEGs decreases in many normalization methods. Consider, for example, comparing DEGs identified by microarray only, RNA-Seq only, and combined (50/50) dataset, and see if it improves enrichment of known signaling pathways.

We greatly appreciate the reviewer's clever suggestion to include signaling pathways as a downstream application. More generally, it led us to reconsider our approach to unsupervised learning. Specifically, we added PLIER analysis using the exact framework described by the reviewer by first analyzing single platform data and then combining platforms to double the sample size, resulting in improved performance.

Pathway-level information extractor (PLIER) (Mao, et al. Nature Methods) starts with a set of known pathways and identifies significant gene expression patterns, resulting in a list of significant pathways. We ran PLIER repeatedly (10x) on data from single platforms (array, RNA-seq) and then on normalized data combining the two platforms (doubling sample size). We report the proportion of pathways found significant, with higher

proportions indicating better performance and ability to detect coordinated biological pathways (please see our new Figure 5 below). Our results show there is a performance enhancement after the increase in sample size (lines 391-407):

3.2.1 Downstream analysis of pathways regulating gene expression

We ran PLIER to identify pathways significantly associated with at least one latent variable in gene expression data derived from a single platform (microarray only or RNA-seq only) or mixed-platform (combination of microarray and RNA-seq). Here, we wanted to examine the benefit of creating a larger data set through the combination of platforms for downstream analysis. For BRCA, each single platform data comprised 174 samples (half of available training samples), while the combined platforms each had 348 samples (the full set of training samples, or, double the amount used for the single platform). For GBM, the sample sizes were 51 for single platform and 102 for combined platforms. In both cancer types, doubling the sample size of our data sets results in a greater proportion of available pathways being significantly associated with an underlying latent variable in the decomposed data set, indicating a greater biologically-relevant signal being extracted from the data (Fig 5). For GBM, with its smaller sample size overall, the proportional benefit of doubling the number of samples was greater than the benefit seen in BRCA. Combined platform data normalized by NPN showed the highest proportion of significant pathways, suggesting that input data closest to a normal distribution on a gene level may be best suited for PLIER. Untransformed data (UN), with its widely differing ranges for array and RNA-seq data, could not be successfully run with PLIER.

Figure 5. Pathways regulating gene expression identified by PLIER. Pathway-level information extractor (PLIER) analysis results showing the proportion of pathways associated with at least one latent variable for BRCA (A) and GBM (B). On the left, Single Platform data represents half the available samples for each platform. On the right, Combined Array and RNA-seq data represents the full set of available samples combined and normalized by various methods. Combined data sets contain twice as many samples as the Single Platform data. (LOG - log₂-transformed; NPN – nonparanormal normalization; QN – quantile normalization; QN-Z – quantile normalization followed by z-scoring; TDM – Training Distribution Matching; Z – z-scoring)

(2) In order to combine unmatched samples from the two platforms, it may be important to find a nonlinear mapping function between microarray probe intensities and RNA-Seq read counts. A good mapping function (or transformation) would project data points from both platforms onto a new manifold such that the dynamic range is maximized. Since the authors already have samples that are measured on both platforms, I recommend visualizing data points in scatter plots (intensity vs read count in log scale) and see how it changes after each normalization. For example, please see Figure 2 of Zhao et al PLoS ONE 2014 doi:10.1371/journal.pone.0078644.

We thank the reviewer for this suggestion and reference to Zhao et al. We performed additional analyses to explore how our normalization methods impact the expression values of matched samples from our hold out data. Taking inspiration from Figure 2 of Zhao et al., we plotted array and RNA-seq expression values from matched samples after each normalization method was applied independently to each set. Please see below for representative scatter plots (hexplots) showing the relationships between array and RNA-seq test data post-normalization for each normalization method.

In the interest of clarity, we note that our experimental design is consistent with how we expect these normalization methods would be used in practice with unmatched samples – that is, array samples in the training set are not matched to RNA-seq samples in the test set. Instead, our experimental design maintains separation between the training set and test set patients from the beginning. Within the training set, the array and RNA-seq samples from each patient are matched. Similarly, array and RNA-seq samples are matched within the test set. When titrating RNA-seq samples into each training set at different proportions, the corresponding matched array sample is removed to maintain a steady number of patients and keep the identity of patients the same.

(3) Test and/or speculate on which normalization method would be suitable for eQTL analysis.

We find the reviewer's question to be compelling and an interesting thought experiment to explore. Yang, et al. found trimmed mean of M-values (TMM) to be the optimal single normalization method for eQTL analysis of RNA-seq data (<https://doi.org/10.1093/bib/bbab193>). However, this does not address the question of cross-platform normalization. One way to approach eQTL analysis is through linear modeling to identify loci where samples with a variant show higher or lower expression of a gene relative to samples with no variation. In that sense, normalization methods that best preserve relative ordering of gene expression values across samples and also maintain some level of normality may be best suited in the eQTL paradigm. However, this initial speculation is untested as eQTL analysis remains an interesting topic for future analysis.

(4) If we transform read counts such that the distribution of normalized microarray intensities does not change, all the available microarray analysis pipelines should simply work fine. Isn't QN (and other well-performing normalizer) somewhat close to doing that?

We appreciate this inquiry as it relates to the motivating question for our manuscript: namely, of the available strategies, which ones appear reasonable for cross-platform normalization. In addition to gaining access to a broader set of microarray analysis tools, by harmonizing RNA-seq to microarray data, we gain the benefit of larger sample sizes required for more data-intensive machine learning applications. But, while gene expression distributions can be adjusted, the particular properties of count-based data, especially with respect to ties and true zero values, may hinder our ability to measure performance across multiple tasks. We have incorporated this valuable perspective and caveat into the discussion section (lines 517-523):

In addition, we stress that the biological question at hand when performing cross-platform normalization must be taken into account, as some assumptions underlying a normalization method may be violated. By harmonizing RNA-seq to microarray data, we gain the benefit of larger sample sizes required for more data-intensive machine learning applications. But, while gene expression distributions can be adjusted, the particular properties of count-based data, especially with respect to ties and true zero values, may hinder our ability to measure performance across multiple tasks.

[Minor points: to improve readability]

(1) Typos: p.4 "affecte" -> "affects", p.11 "that that" -> "that"

Thank you -- we have fixed these errors.

(2) Please describe in more detail what the rationale is behind creating "reconstructed" holdout datasets and then fitting classifiers in PCA/ICA tests. I found it was complex and a bit difficult to understand.

We have incorporated the reviewer's experience and added more rationale behind our process of dimensionality reduction, reconstruction, and subtype prediction. Many existing tools utilize dimensionally-reduced expression data, whether from principal component analysis (PCA) or another method (Way, et al.). (Note: we have removed independent component analysis from our methods due to redundancy.) We hypothesized that normalization methods may distort the utility of low-dimensional projections and that this distortion may differ by the level of RNA-seq data titrated into each training set. Our purpose in including PCA reconstruction of the test data and subsequent subtype prediction was to evaluate the ability of machine learning tasks to work after projecting the test data onto the low-dimensional space defined by the training data. We evaluated reconstruction error using the mean absolute scaled error (MASE), and found quantile normalization and training distribution matching methods to have the lowest error and best prediction performance. Please see the rationale we added to the manuscript (lines 281-285):

Principal Components Reconstruction for Subtype Prediction: We hypothesized that normalization methods may distort the utility of low-dimensional projections and that this distortion may differ by the level of RNA-seq data titrated into each training set. To test this, we evaluated machine learning tasks after projecting the test data onto the low-dimensional space defined by the training data.

(3) Please describe what NPN does in more detail so that readers do not have to look up the reference to it. How is it different from Z or rank-based inverse normal transformations as in Beasley et al 2009 doi:10.1007/s10519-009-9281-0?

We thank the reviewer for this question and opportunity to include additional methods detail in the manuscript. We applied nonparanormal (NPN) transformation using the implementation in the R package huge (<https://CRAN.R-project.org/package=huge>). To use Beasley et al.'s terms, NPN is a deterministic, fractional, rank-based inverse normal transformation performed on the gene level. We used the "shrinkage" empirical cumulative distribution function, which does not winsorize the extreme tail values like the "truncation" eCDF does. With "shrinkage", the inverse normal transformation is applied to scaled rank values. Rank values are scaled down by a factor of the number of genes + 1. We have updated our methods section to provide additional details to readers (lines 180-183):

Non-paranormal normalization (NPN): NPN forces data to conform to a normal distribution by rank-transformation followed by quantile normalization, placing each observation where it would fall on a normal distribution. NPN was

performed using the huge R package prior to concatenating samples from both platforms and separately on single-platform holdout sets.

(4) Provide formula for kappa statistics.

We have included a formula for kappa in section 2.2.4 as well as additional context for why the Kappa metric was chosen. Please see our updated manuscript text (lines 243-254):

We used the Kappa statistic to evaluate performance on holdout data for two main reasons. The first reason is we make no assumptions about class balance in our data, so to mitigate the potential for bias due to class imbalance, we chose a metric that builds in a baseline probability of chance agreement. The second reason is to accommodate our multi-class outcome (e.g., five subtypes in both breast cancer and glioblastoma), which Kappa achieves without relying on a composite of one-vs-all comparisons, as classes were not balanced. Briefly, the Kappa statistic takes into account the expected accuracy of a random classifier and is generally considered to be less misleading than observed accuracy alone. The formula for Cohen's Kappa is:

$$\kappa = \frac{p_0 - p_e}{1 - p_e}$$

where p_0 is the agreement observed between two methods and p_e is the expected probability of agreement by chance.

(5) [Section 3.2] " We found that projecting the holdout sets onto QN and TDM normalized training space results in less loss of subtype classifier performance": Please rephrase using a quantitative metric.

We thank the reviewer for seeking quantification of our claim. We have edited the section to now include a quantitative comparison between Kappa values from prediction on non-reconstructed test data and reconstructed test data in a new supplementary figure that visualizes the differences (see Figure S7 below) (lines 423-425 and 428-430):

In general, the random forest classifier suffered the largest loss of performance, likely due to gene expression thresholds (rules) used for prediction (Fig S7).

and

We found that projecting the holdout sets onto QN and TDM normalized training space results in less loss of subtype classifier performance relative to when there is no reconstruction performed (Fig S7).

Supplementary Figure 7. Difference in subtype classifier Kappa values between non-reconstructed and reconstructed microarray and RNA-seq test data from (A) BRCA and (B) GBM. Line plots of the median difference in Kappa statistics from 10 repeats of steps 1-3A and 3B from Figure 1 and for six normalization methods (and untransformed data) are displayed. Median values are shown as points. (LOG - log₂-transformed; NPN - nonparanormal normalization; QN - quantile normalization; QN-Z - quantile normalization followed by z-score; TDM - Training Distribution Matching; UN - untransformed; Z - z-score)

(6) [Figure S1] Consider showing composition rather than counts if your goal is to show that the subtype distribution are close to each other.

We have carefully considered the reviewer's suggestion, leading us to include additional information in the original plots to quantify the proportions of each category present, not just the count. Please see our updated Figure S1 below:

Supplementary Figure 1. Predictor class balance in the whole set (all samples) , training set (two-thirds), and test set (one-third). (A) BRCA subtype. (B) GBM subtype. (C) BRCA TP53 mutation status. (D) GBM TP53 mutation status. (E) BRCA PIK3CA mutation status. (F) GBM PIK3CA mutation status.

(7) [Figure 3, S4, and S5] Put y-axis in a same scale on the same set of grids so readers can directly compare between subplots.

We appreciate the reviewer's concern that subplots were not easily compared due to issues of scale. As much as possible, we have implemented this important change to the presentation of our data across the entire project. For example, please see our updated Figure 3 below:

Figure 3. BRCA subtype classifier performance on microarray and RNA-seq test data. (A) Violin plots of Kappa statistics from 10 repeats of steps 1-3A from Figure 1 and for six normalization methods (and untransformed data) are displayed. Median values are shown as points. (LOG - log₂-transformed; NPN - nonparanormal normalization; QN - quantile normalization; QN-Z - quantile normalization followed by z-score; TDM - Training Distribution Matching; UN - untransformed; Z - z-score)

(8) Specify what reference is used for LOG normalizing microarray holdout set.

We have improved the documentation of our normalization methods, including a more thorough explanation of the process of log normalization in our pipeline. There is no reference used to LOG normalize the microarray test data. Holdout microarray data is LOG transformed using $\log_2(x+1)$ and is then rescaled to [0-1] on the gene level. We have included a new supplementary figure that details how each part of our training and test data sets are normalized. Please see our new Figure S2 (below) and updated methods description (lines 219-222):

Microarray test data (100% microarray): Similar to the 100% array training data, array test data gets LOG, NPN, QN, QN-Z, UN, and Z transformed by itself without reference to any other data, and it then undergoes gene-level zero-to-one transformation after each normalization except UN. There is no TDM for 100% array test data.

Supplementary Figure 2. Overview of cross-platform normalization with and without reference. Double hatched boxes refer to data processed using microarray data as a reference distribution. Single hatched boxes refer to data processed without using microarray data as a reference (no reference). Orange boxes refer to microarray data. Blue boxes refer to RNA-seq data. (A) Matched samples are split into a training set and testing set. (B) Each training set is composed of between 0% and 100% RNA-seq data. For No Reference, the microarray portion and RNA-seq portion are normalized separately for LOG, NPN, and Z. With Reference, the microarray portion remains unprocessed (LOG) while the RNA-seq portion is normalized using the microarray portion as a reference distribution for QN, QN-Z, and TDM. Microarray and RNA-seq portions are then combined to form each cross-platform normalized training data set. (C) Microarray test data normalization occurs without reference for LOG, NPN, QN, QN-Z, and Z, and without reference for RNA-seq data for LOG, NPN, and Z. RNA-seq test data normalization uses microarray training data as the reference distribution for QN, QN-Z, and TDM at each percentage of RNA-seq titration in training. (LOG - log₂-transformed; NPN - nonparanormal normalization; QN - quantile normalization; QN-Z - quantile normalization followed by z-score; TDM - Training Distribution Matching; UN - untransformed; Z - z-score)

(9) When RNA-Seq samples are titrated, are the corresponding samples excluded from the microarray samples?

Yes -- when RNA-seq samples are titrated, the corresponding array samples are removed. This keeps the number of samples consistent across titration levels as well as maintains the category distribution (subtype or gene mutation status) of the patients present in the cohort. We have edited the methods section to ensure this important point is communicated clearly (lines 111-121):

2.2.1 Experimental design

An overview of our experimental design for machine learning evaluations is illustrated in Fig 1. Matched samples were split into training (2/3) and test (1/3) sets using the `createDataPartition` function in the `caret` package, which takes the balance of the class distributions in the training and holdout sets into account (i.e., stratified sampling; Fig 1A). See Fig S1 for representative plots of subtype and mutation status distributions for BRCA and GBM. To create our panel of training sets, samples analyzed with RNA-seq were "titrated" into the training set via random selection in 10% increments to produce training sets containing 0%, 10%, 20% ... 100% RNA-seq data (Fig 1B). For every RNA-seq sample added to a training set, the matched microarray sample was removed, keeping the number and identity of patients consistent across all training data sets.

Reviewers' comments:

Reviewer #1 (Remarks to the Author):

This paper took the authors five years to revise. Over the years, many alignment/integration methods have been developed in the single-cell field. Hence, these cutting-edge alignment methods may be applicable to the task in this paper, and the authors should add a comparison and discussion.

A fundamental issue with the DE analysis is that the definition of DE genes depends on the data scale. In other words, the definition is not scale invariant. The mathematical reason behind this is that DE analysis tests the equality of two gene's expected expression levels, and this equality may no longer hold after the expression levels are non-linearly transformed. That is, $E[X] = E[Y]$ does not imply $E[f(X)] = E[f(Y)]$ if f is non-linear. Hence, the DE results are uninformative.

For the PLIER pathway analysis, finding more significant pathways is not necessarily better. The authors need to provide a more detailed analysis to justify why pathways found from the combined data are more reasonable than the pathways found from the single-platform data.

In MASE, why use the absolute error instead of the squared error?

Figure 6A is not conclusive and thus not informative.

The authors did not directly answer Reviewer 1's Comment 7, which is about why RNA-seq cannot be used as a reference distribution to normalize microarray data.

Response to Reviewer 3's Comment (2) is not informative. What is to be learned from the hexplots?

How is NPN different from the inverse normal transformation? The author's reply to Reviewer 3's minor comment (3) is not informative.

Reviewer #2 (Remarks to the Author):

The revised paper addresses the comments from my original review. I am especially impressed by the authors' use of Docker to solve the software issues.

The paper presents a nice empirical evaluation of different cross-platform normalization methods for a broad range of different analysis tasks. The paper is well-written and clear, and its fundamental contribution is interesting and worth publishing at Communications Biology. I have two remaining major questions I believe should be addressed before the paper can be considered for publication, though:

1. The headline result, "differential expression analysis is possible on mixed platform data sets" seems overly optimistic to me. One critical factor of differential expression analysis is formal statistical significance testing with FDR control. The authors make no effort to estimate FDR for their combined results, and given the low similarity in some cases, it seems likely this would be nowhere near advertised level. To fix this, the authors should either extend their method with more formal statistical analysis, or edit their claims to make it clear no formal statistical guarantees can be given for the results from mixed data.

2. In evaluation of significant pathways (Fig. 5), the evaluation appears to focus only on the number of identified pathways, completely ignoring the issue of potential false positives. The evaluation would need to be redone in a way that differentiates between true positives (pathways identified from full array or RNA-seq data, or even better from both) and false positives (others) instead of just counting the number of discoveries.

Reviewer #4 (Remarks to the Author):

This is an interesting work examining the normalization methods on cross-platform gene expression data for machine learning and differential expression analysis. The main concerns are listed below,

- (1) It is a baseline comparison work for normalization methods of expression data. The authors did not develop any normalization method combining microarray and RNA-seq data, so the title "Cross-platform normalization enables machine learning model training on microarray and RNA-seq data simultaneously" does not fill the content.
- (2) The authors restricted the data set to the tumor samples that were measured on both platforms. How about the common situation when no tumor samples measured on the two platforms? Or in other words, how about combining two breast cancer datasets from different platforms based on the GEO database?
- (3) Again, the authors claimed 'it is possible to perform effective cross-platform normalization and combine microarray and RNA-seq data for machine learning applications. ROC or PRC is important for evaluating the performance. On top of that, sensitivity and specificity based on a union threshold across different datasets are recommended.
- (4) Other normalization methods like CrossNorm should also be included for the baseline comparison.
- (5) RNA-seq data or microarray data are also detected using a series of platforms, the authors should discuss the effect between different platforms (such as Affy, agilent and beats) and different techniques (RNA-seq and array).

Reviewer #1 (Remarks to the Author):

We thank the reviewer for the time and effort they have invested in improving this manuscript. We have responded to each of the reviewer's important points below and have made several key changes to the manuscript to address their concerns.

This paper took the authors five years to revise. Over the years, many alignment/integration methods have been developed in the single-cell field. Hence, these cutting-edge alignment methods may be applicable to the task in this paper, and the authors should add a comparison and discussion.

We were excited for the reviewer's suggestion and eager to see if recent developments from the single cell field could apply to our research question. Namely, could methods optimized for single cell integration accurately reshape array and RNA-seq data to allow for a joint data analysis? We settled on the integration methods pipeline from the widely-used single cell R package Seurat. Seurat's integration algorithm identifies anchor points in each data set and seeks to transform each dataset to match the other in a reduced dimension space. The analogy connecting single cell integration with bulk normalization was that single samples acted as single cells, and the two platforms of array and RNA-seq data acted as two batches of data needing integration.

We attempted to integrate our array and RNA-seq training data according to our titration protocol, ranging from data with 10% RNA-seq to data with 90% RNA-seq. We ultimately found we had insufficient sample sizes at the lower and upper ends of the titration range to allow for a full comparison across the titration spectrum as with other methods. Given this limitation, we have made results available in table form along with the results from other methods (see Data Availability) rather than include incomplete Seurat results in our display figures. We added the following methods description to the manuscript (lines 217-224):

Other approaches to combining and normalizing gene expression data are commonly used in single cell RNA-sequencing studies to integrate data sets by reducing batch effects across studies and experimental conditions. We designed an approach analogous to Seurat's integration pipeline in which single cells became single (bulk) samples, and our array and RNA-seq datasets were the two batches or experimental conditions. Due to low sample numbers at the edges of our titration protocol, many experimental conditions could not be integrated. For those conditions with enough samples from array and RNA-seq for successful integration, we have included machine learning prediction results in table form (see Data Availability).

A fundamental issue with the DE analysis is that the definition of DE genes depends on the data scale. In other words, the definition is not scale invariant. The mathematical reason behind this is that DE analysis tests the equality of two gene's expected expression levels, and this equality

may no longer hold after the expression levels are non-linearly transformed. That is, $E[X] = E[Y]$ does not imply $E[f(X)] = E[f(Y)]$ if f is non-linear. Hence, the DE results are uninformative.

Thank you – we agree with the reviewer's concern and have elected to remove our differential expression results from the manuscript. Our text, main figures, and supplementary figures have been updated to reflect this change.

For the PLIER pathway analysis, finding more significant pathways is not necessarily better. The authors need to provide a more detailed analysis to justify why pathways found from the combined data are more reasonable than the pathways found from the single-platform data.

We thank the reviewer for suggesting this opportunity for a deeper analysis of our PLIER pathway results. We found that our ability to detect significant signals from gene expression data using the PLIER method depended on the sample size of the input, consistent with previous reports (Taroni JN, Grayson PC, Hu Q, Eddy S, Kretzler M, Merkel PA, et al. MultiPLIER: a transfer learning framework for transcriptomics reveals systemic features of rare disease. *Cell Systems*. 2018;8: 395947.). This makes sense because increasing sample size increases PLIER's power to detect correlated expression patterns in the training set, some of which are expected to be associated with biological pathways. We appreciate the reviewer's concern that increasing sample size may also increase the opportunity for spurious connections to be found (i.e., increased false positives rate). Thus, we sought to contextualize the rate of significant results returned by PLIER at different sample sizes by running a negative control experiment to establish the baseline false positive rate of significant results obtained by chance. By permuting the relationship between genes and pathways and running PLIER as before, we can understand how often pathways are reported as significant when there is no biological reason for it.

To accomplish this, we permuted the gene (rows) and pathways (columns) matrix by resampling without replacement within each column such that the number of genes associated with each pathway remained constant. We re-ran PLIER using the permuted gene x pathway matrix 10 times for each dataset, including single platform data at half size and full size and full size cross-platform normalized data, for a total of 110 permuted PLIER runs. We found that, when the biological relationships represented in the gene-pathway input matrix are permuted, no or very few pathways are returned as significant. Specifically, out of 110 permuted PLIER runs, 105 reported no significant pathways, while 5 reported a single significant pathway, for a maximum observed false positive rate of 1 out of 817 pathways, or 0.12%. This rate is much lower than the typical range of 10-30% observed from BRCA and GBM data processed without permutation, so we have confidence that false positives account for only a small percentage of the observed rates of significant pathways.

We also updated our study design to include the full set of samples from each platform and found a similar proportion of pathways were returned as significant for

cross-platform and single-platform training datasets of equal sample size. Please see the updated figure below.

We have updated the methods and results sections to include this new baseline false positive rate and full-size single platform study design, and we have included the permuted gene-relationship results with our other available data.

Methods (lines 314-323):

As a negative control, we identified the baseline false positive rate of significant results obtained by chance by permuting the gene-pathway relationships represented in the gene-pathway input matrix. Specifically, we permuted the gene (rows) and pathways (columns) matrix by resampling without replacement within each column such that the number of genes associated with each pathway remained constant. We compared results from single platform datasets (microarray or RNA-seq only) with normalized data comprising 50% microarray and 50% RNA-seq samples. For the single platform datasets, we included some with half of the available samples and some with the full set of samples. The normalized datasets comprised the full set of samples with half coming from either platform.

Results (lines 468-483):

We ran PLIER to identify pathways significantly associated with at least one latent variable in gene expression data derived from a single platform (microarray only or RNA-seq only) or mixed-platform (combination of microarray and RNA-seq). Here, we wanted to examine the benefit of creating a larger data set through the combination of platforms for downstream analysis. For BRCA, the half-size single platform data comprised 174 samples (half of available training samples), while the full-size single platform and cross-platform data each had 348 samples (the full set of training samples). For GBM, the sample sizes were 51 for half-size and 102 for full-size datasets. As a negative control, we permuted gene-pathway relationships used as input to PLIER to establish the baseline rate of false positive pathway associations. We found that out of 110 PLIER runs, five runs reported one false positive significant pathway association, while 105 runs reported no significant pathways.

Overall, in both cancer types, doubling the sample size of our data sets resulted in a greater proportion of available pathways being significantly associated with an underlying latent variable in the decomposed data set, with a similar increase observed for both cross-platform and single platform full-size datasets. This could indicate a greater biologically-relevant signal being extracted from the data (Fig 4A-B) and is consistent with previous studies.

Figure 4. Pathways regulating gene expression identified by PLIER. Pathway-level information extractor (PLIER) analysis results showing the proportion of pathways associated with at least one latent variable for BRCA (A) and GBM (B). The left panel shows single platform data using half the available samples for each platform. The middle panel shows 50% array and 50% RNA-seq data combined and normalized by various methods representing the full set of available samples. The right panel shows single platform data using the full set of available samples for each platform. (LOG - log₂-transformed; NPN – nonparanormal normalization; QN – quantile normalization; QN (CN) – quantile normalization with CrossNorm; QN-Z – quantile normalization followed by z-scoring; TDM – Training Distribution Matching; Z – z-scoring)

In MASE, why use the absolute error instead of the squared error?

We appreciate the reviewer's question regarding methodological detail. We used MASE (mean absolute scaled error) in the context of reconstruction error: after defining a PCA transformation, how similar are the original data and the original data projected into PCA space? We chose MASE for its attributes, in particular: scale invariance, symmetrical treatment of over and under prediction, and its behavior close to 0. Any error measure that incorporates those attributes would be appropriate here, including a modified version of MASE that uses squared error instead of absolute error.

Possible advantages of using a squared error MASE would be heavier penalization of more inaccurate predictions. Advantages of using absolute error MASE are that it is less subject to outliers (a single large outlier could unfairly skew results if the error was squared) and absolute errors are more interpretable since they maintain a linear (rather than quadratic) relationship with the original scale.

Figure 6A is not conclusive and thus not informative.

Figure 6A refers to differential expression results and, based on other advice taken from the reviewer, we have chosen to remove it from the manuscript. Our text, main figures, and supplementary figures have been updated to reflect this change.

The authors did not directly answer Reviewer 1's Comment 7, which is about why RNA-seq cannot be used as a reference distribution to normalize microarray data.

We appreciate the reviewer's question seeking clarification on why RNA-seq cannot be used as the reference distribution to which microarray data is normalized. One of the benefits of RNA-seq over array data is its broader dynamic range. Implicit within that description is the notion that RNA-seq better captures the true abundance of transcripts than its array-based predecessors. When normalizing these two data types to the same distribution, it is necessary to reshape the broader and more precise data to fit the narrower and less precise distribution. By analogy, the points on a clear photograph can be mapped onto a pixelated version of the same image, but not the reverse. We have added this perspective to the introduction (lines 66-68) as well as our github page (https://github.com/greenelab/RNAseq_titration_results).

Given the broader dynamic range of RNA-seq data, this work focuses on the problem of normalizing RNA-seq data to a target distribution of array data.

Response to Reviewer 3's Comment (2) is not informative. What is to be learned from the hexplots?

In a previous round of review, we followed Reviewer 3's suggestion that "recommend[ed] visualizing data points in scatter plots (intensity vs read count in log scale) and see how it changes after each normalization." They referenced Figure 2 of Zhao et al PLoS ONE 2014 doi:10.1371/journal.pone.0078644 as an example. We interpreted the suggestion to be an exploration of this two-fold question: how does normalization impact the shape of the data, and how do the array and RNA-seq values compare after normalization? We sought to emulate the spirit of the graphs shown in Figure 2 but with a different style (hex plot) that avoids overplotting of data points. We generated the plots using paired array and RNA-seq test samples from our BRCA test cohort (n = 172 pairs of samples). For a random subset of 10K genes, we plotted the x-y coordinates of the paired expression values. The overall trend of the data is reassuring in that array and RNA-seq values are comparable, and the different shapes of the point clouds illustrate how the data changes due to each normalization method. For example, the NPN hex plot shows the normalized data to have a well-defined Gaussian shape. Please see the visualization below.

How is NPN different from the inverse normal transformation? The author's reply to Reviewer 3's minor comment (3) is not informative.

We appreciate the opportunity to clarify and expand upon our previous response given to Reviewer 3. Specifically, we can expand upon the discussion of how our implementation of the nonparanormal normalization (NPN) does fit within the framework of rank-based inverse normal transformations.

Beasley, *et. al* (doi:10.1007/s10519-009-9281-0) describe a general rank-based inverse normal transformation (INT) in the following way:

Perhaps the most commonly used rank-based INT transformation entails creating a modified rank variable and then computing a new transformed value of the phenotype for the i th subject:

$$Y_i^t = \Phi^{-1}\left(\frac{r_i - c}{N - 2c + 1}\right)$$

where r_i is the ordinary rank of the i th case among the N observations and Φ^{-1} denotes the standard normal quantile (or probit) function.

In our implementation of NPN, we relied on the `huge.npn()` function from the R package `huge` (<https://cran.r-project.org/web/packages/huge/index.html>). With the default option for `npn.func = "shrinkage"`, `huge.npn()` applies the `qnorm()` function to ranks of values within each column assuming the correction value c from the formula above is 0. After inverse normalization, `npn.huge()` then scales the entire resulting matrix by the standard deviation of the first column, which approaches 1 as N grows. Other `npn.func` options given by `huge.npn()` also fit within the INT framework but with modifications like, for example, winsorizing values above or below a given threshold.

We have updated the relevant section of text to make this clear (lines 203-207):

Non-paranormal normalization (NPN): Our implementation of NPN is a rank-based inverse normal transformation that forces data to conform to a normal distribution by rank-transformation followed by quantile normalization, placing each observation where it would fall on a standard normal distribution. NPN was performed using the huge R package prior to concatenating samples from both platforms and separately on single-platform holdout sets.

Reviewer #2 (Remarks to the Author):

The revised paper addresses the comments from my original review. I am especially impressed by the authors' use of Docker to solve the software issues.

The paper presents a nice empirical evaluation of different cross-platform normalization methods for a broad range of different analysis tasks. The paper is well-written and clear, and its fundamental contribution is interesting and worth publishing at Communications Biology.

We thank the reviewer for their affirming review, and we appreciate their recognition of our efforts to enhance reproducibility and software versioning through Docker. We have followed the reviewer's feedback on the remaining questions which have strengthened the manuscript. In particular, we decided to remove our differentially expressed gene sections in order to avoid overstating our findings. We have also improved our expression pathway evaluation by establishing a negative control false positive rate through permutation of the gene-pathway relationships and we updated our study design to include single platform full-size datasets as another comparison for our cross-platform normalized data.

I have two remaining major questions I believe should be addressed before the paper can be considered for publication, though:

1. The headline result, "differential expression analysis is possible on mixed platform data sets" seems overly optimistic to me. One critical factor of differential expression analysis is formal statistical significance testing with FDR control. The authors make no effort to estimate FDR for their combined results, and given the low similarity in some cases, it seems likely this would be nowhere near advertised level. To fix this, the authors should either extend their method with more formal statistical analysis, or edit their claims to make it clear no formal statistical guarantees can be given for the results from mixed data.

We agree with the reviewer's concern regarding the differentially expressed genes aspect of our manuscript. In order to keep the focus of our analysis on supervised machine learning and downstream pathway analysis (like PLIER), we elected to remove our DEG sections and figures. Our text, main figures, and supplementary figures have been updated to reflect this change.

2. In evaluation of significant pathways (Fig. 5), the evaluation appears to focus only on the number of identified pathways, completely ignoring the issue of potential false positives. The evaluation would need to be redone in a way that differentiates between true positives (pathways identified from full array or RNA-seq data, or even better from both) and false positives (others) instead of just counting the number of discoveries.

We thank the reviewer for their suggestion to include full array and RNA-seq data in our comparison of the significant pathways returned by PLIER. This insight fit nicely with our additional analysis establishing a baseline false positive rate. Based on previous studies,

we expected we would detect more significant signals from gene expression data using PLIER with greater sample size (Taroni JN, Grayson PC, Hu Q, Eddy S, Kretzler M, Merkel PA, et al. MultiPLIER: a transfer learning framework for transcriptomics reveals systemic features of rare disease. Cell Systems. 2018;8: 395947.). Adding more samples increases PLIER's power to detect correlated expression patterns in the training set, some of which are expected to be associated with biological pathways. The other issue at play is that increasing sample size may also increase the false positive rate. We were able to contextualize the rate of significant results returned by PLIER at different sample sizes by running a negative control experiment to establish the baseline false positive rate of significant results obtained by chance. By permuting the relationship between genes and pathways and running PLIER as before, we can understand how often pathways are reported as significant when there is no biological reason for it.

To accomplish this, we permuted the gene (rows) and pathways (columns) matrix by resampling without replacement within each column such that the number of genes associated with each pathway remained constant. We re-ran PLIER using the permuted gene x pathway matrix 10 times for each dataset, including single platform data at half size and full size and full size cross-platform normalized data, for a total of 110 permuted PLIER runs. We found that, when the biological relationships represented in the gene-pathway input matrix are permuted, no or very few pathways are returned as significant. Specifically, out of 110 permuted PLIER runs, 105 reported no significant pathways, while 5 reported a single significant pathway, for a maximum observed false positive rate of 1 out of 817 pathways, or 0.12%. This rate is much lower than the typical range of 10-30% observed from BRCA and GBM data processed without permutation, so we have confidence that false positives account for only a small percentage of the observed rates of significant pathways.

We also updated our study design to include the full set of samples from each platform and found a similar proportion of pathways were returned as significant for cross-platform and single-platform training datasets of equal sample size. Please see the updated figure below.

We have updated the methods and results sections to include this new baseline false positive rate and full-size single platform study design, and we have included the permuted gene-relationship results with our other available data.

Methods (lines 314-323):

As a negative control, we identified the baseline false positive rate of significant results obtained by chance by permuting the gene-pathway relationships represented in the gene-pathway input matrix. Specifically, we permuted the gene (rows) and pathways (columns) matrix by resampling without replacement within each column such that the number of genes associated with each pathway remained constant. We compared results from single platform datasets

(microarray or RNA-seq only) with normalized data comprising 50% microarray and 50% RNA-seq samples. For the single platform datasets, we included some with half of the available samples and some with the full set of samples. The normalized datasets comprised the full set of samples with half coming from either platform.

Results (lines 468-483):

We ran PLIER to identify pathways significantly associated with at least one latent variable in gene expression data derived from a single platform (microarray only or RNA-seq only) or mixed-platform (combination of microarray and RNA-seq). Here, we wanted to examine the benefit of creating a larger data set through the combination of platforms for downstream analysis. For BRCA, the half-size single platform data comprised 174 samples (half of available training samples), while the full-size single platform and cross-platform data each had 348 samples (the full set of training samples). For GBM, the sample sizes were 51 for half-size and 102 for full-size datasets. As a negative control, we permuted gene-pathway relationships used as input to PLIER to establish the baseline rate of false positive pathway associations. We found that out of 110 PLIER runs, five runs reported one false positive significant pathway association, while 105 runs reported no significant pathways.

Overall, in both cancer types, doubling the sample size of our data sets resulted in a greater proportion of available pathways being significantly associated with an underlying latent variable in the decomposed data set, with a similar increase observed for both cross-platform and single platform full-size datasets. This could indicate a greater biologically-relevant signal being extracted from the data (Fig 4A-B) and is consistent with previous studies.

Figure 4. Pathways regulating gene expression identified by PLIER. Pathway-level information extractor (PLIER) analysis results showing the proportion of pathways associated with at least one latent variable for BRCA (A) and GBM (B). The left panel shows single platform data using half the available samples for each platform. The middle panel shows 50% array and 50% RNA-seq data combined and normalized by various methods representing the full set of available samples. The right panel shows single platform data using the full set of available samples for each platform. (LOG - log₂-transformed; NPN - nonparanormal normalization; QN - quantile normalization; QN (CN) - quantile normalization with CrossNorm; QN-Z - quantile normalization followed by z-scoring; TDM - Training Distribution Matching; Z - z-scoring)

Reviewer #4 (Remarks to the Author):

This is an interesting work examining the normalization methods on cross-platform gene expression data for machine learning and differential expression analysis.

We greatly appreciate the reviewer's contribution to improving our manuscript. In addressing the questions raised below, we have followed the reviewer's guidance by including additional performance metrics, adding another normalization method (CrossNorm) for comparison, and ensuring our study design and methods descriptions are clear and accurately match our claims.

The main concerns are listed below,

(1) It is a baseline comparison work for normalization methods of expression data. The authors did not develop any normalization method combining microarray and RNA-seq data, so the title "Cross-platform normalization enables machine learning model training on microarray and RNA-seq data simultaneously" does not fill the content.

We appreciate the reviewer's concern that our title may not accurately reflect the work presented in the manuscript. In our work, we sought to determine if existing normalization methods adequately combine array and RNA-seq data in a way that makes machine learning possible. As the reviewer correctly notes, we did not develop new normalization methods to combine microarray and RNA-seq data. Rather, our study design implemented existing normalization methods and showed how those methods allow, or enable, machine learning modeling of array and RNA-seq platforms together.

After carefully reconsidering our title, our opinion is that our word choice accurately reflects the content of our manuscript. We have taken care to reemphasize the nature of our work both in the abstract (lines 17-24):

Here we perform supervised and unsupervised machine learning evaluations to assess which existing normalization methods are best suited for combining microarray and RNA-seq data...

We demonstrate that it is possible to perform effective cross-platform normalization using existing methods to combine microarray and RNA-seq data for machine learning applications.

and in the main text (lines 78-80):

We aimed to assess the extent to which it was possible to effectively normalize and combine microarray and RNA-seq data with existing methods for use as a training set for machine learning applications.

(2) The authors restricted the data set to the tumor samples that were measured on both platforms. How about the common situation when no tumor samples measured on the two platforms? Or in other words, how about combining two breast cancer datasets from different platforms based on the GEO database?

We thank the reviewer for pointing out this alternative study design, which we understand to mean combining data from array-based studies with data produced using different array kits from different companies, like Agilent or Affymetrix, or combining different versions of kits from the same company.

The spirit of that array-based design closely matches our study design which specifically addresses the case where researchers wish to normalize and combine expression data from array-based and sequencing-based datasets. We view this unmet need of array and RNA-seq normalization as a more challenging opportunity for dataset integration. This is especially true given the acceleration of publicly-available RNA-seq data in the past five years compared to array data. Our view is that the methods used in this work to normalize RNA-seq and array data together would also apply to normalizing two different array datasets together.

In this study, our machine learning experimental study design combining array and RNA-seq required that each patient have matched data from both array and RNA-seq platforms to ensure any differences we saw were due to the technology and not sample composition. We would have had the same matched sample requirement if we were assessing normalization methods to combine different array-based platforms. In practical application, the two datasets would not be bound by this matching constraint, but our experimental evaluation of normalization methods required it.

We have added the following exploration of this connection between study designs into the discussion section (lines 534-536):

Other applications of cross-platform normalization, such as combining array-based gene expression data from different array manufacturers (e.g., Agilent and Affymetrix) could follow the same principles as what we have described here.

(3) Again, the authors claimed 'it is possible to perform effective cross-platform normalization and combine microarray and RNA-seq data for machine learning applications. ROC or PRC is important for evaluating the performance. On top of that, sensitivity and specificity based on a union threshold across different datasets are recommended.

We thank the reviewer for suggesting the addition of ROC or PRC, sensitivity, and specificity to our model metrics. Our first choice of model metric for this study was kappa, which we chose because our outcomes are both unbalanced and multi-class. Kappa avoids bias due to unbalanced data by accounting for the underlying possibility of

obtaining correct predictions by chance. Kappa also avoids performing an unweighted combination of one-vs-rest summary statistics.

We calculated the area under the ROC curve (AUC) for our Lasso and Random Forest models. AUC calculations failed for some SVM models due to how class probabilities are determined for SVM models fit by the kernlab package. We also calculated sensitivity and specificity for each class of model. Kappa, AUC, sensitivity, and specificity metrics for each model are now reported in our new Supplementary Table 1 of the manuscript. We found Kappa, AUC, sensitivity, and specificity to show similar patterns of performance across modeling and normalization methods.

Kappa:

AUC (no AUC values for Linear SVM models):

Sensitivity:

Specificity:

We have included the original (Kappa) and additional (AUC, sensitivity, and specificity) metrics in a newly created Supplementary Table 1 and have added this description to the methods (lines 291-294):

In addition to the Kappa statistic, we used a composite one-vs-all approach to calculate several metrics designed for binary classification tasks with balanced data, including the area under the receiver operator curve (AUC), sensitivity, and specificity. Metrics for all subtype and mutation prediction models may be found in Supplementary Table 1. Metrics for all subtype and mutation prediction models may be found in Supplementary Table 1.

(4) Other normalization methods like CrossNorm should also be included for the baseline comparison.

We appreciate the reviewer's suggestion to add an additional normalization method to our study. We found the CrossNorm method to be an interesting approach and have implemented the general CrossNorm algorithm with quantile normalization using publicly-available code (Cheng L, Lo L-Y, Tang NLS, Wang D, Leung K-S. CrossNorm: a novel normalization strategy for microarray data in cancers. Sci Rep. 2016;6: 18898;

<https://www.nature.com/articles/srep18898>). We found CrossNorm's performance similar to quantile normalization for supervised learning purposes but observed worse performance in the PLIER pathway analysis context. We added the following methods description to our methods section (lines 173-182):

Quantile normalization with CrossNorm (QN (CN)): CrossNorm combines data from different distributions by stacking columns from paired samples or, in the more general case, combinatorially stacking columns from each distribution before performing quantile normalization. Before combining data sets, we array and RNA-seq sample expression values were rescaled zero-to-one. The QN (CN) process differs from QN or QN-Z because RNA-seq samples are not normalized to fit the array distribution; rather, the samples from array and RNA-seq are cross-normalized toward the same shared distribution together. Since CrossNorm requires two data sets, there is no QN (CN) data for training at 0% and 100% RNA-seq. Likewise, QN (CN) array and RNA-seq test data is simply rescaled and quantile normalized without reference to array training data.

(5) RNA-seq data or microarray data are also detected using a serious of platforms, the authors should discuss the effect between different platforms (such as Affy, agilent and beats) and different techniques (RNA-seq and array).

We thank the reviewer for further articulating this alternative study design choice between focusing on RNA-seq and array normalization, which our study does, and an alternative study design with various types of array data. We also note the potential semantic overlap in the use of the word platform. In this study, we use "platform" (as in, cross-platform) to broadly refer to the two main classes of bulk RNA expression measurement: array and RNA-seq. Platform may also have a more specific interpretation when we consider the different implementations of array-based technologies within the array class.

We are encouraged by the relatively similar performances we saw in our two cancer types, given that our BRCA array data were from Agilent while our GBM array data were from Affymetrix. This indicates that both Agilent and Affymetrix array data are suitable for normalization with RNA-seq data. It is our view that the approach used here to combine RNA-seq and array data could also work for normalizing array datasets together. To communicate this, we added this point to the discussion (lines 534-536):

Other applications of cross-platform normalization, such as combining array-based gene expression data from different array manufacturers (e.g., Agilent and Affymetrix) could follow the same principles as what we have described here.

Reviewers' comments:

Reviewer #1 (Remarks to the Author):

1. The authors mentioned in the last paragraph of the Introduction that the normalization approaches are applied to multiple applications: supervised machine learning, unsupervised machine learning, and pathway analysis. However, in the following sections, pathway analysis is discussed in the section on unsupervised machine learning. Therefore, pathway analysis should be part of unsupervised learning instead of a separate application. Furthermore, the author can explain more on the reason for focusing on those supervised and unsupervised machine learning applications.

2. In the Methods section, the authors introduced the LOG transformation. Specifically, "the microarray data was inverse log-transformed and then log-transformed...". Is inverse-transformation same as taking exponential? If so "inverse log-transformed and then log-transformed" does not make any change. The authors should give a clearer definition of the "inverse log-transform".

3. In the introduction of Quantile normalization with CrossNorm, there is a typo in the sentence "we array and RNA-seq sample expression values were rescaled zero-to-one." It should be "the array and RNA-seq sample expression values ...".

4. In the description of Figure 2 and 3, it says "violin plots...". However, the plots are not violin plots.

5. The negative control added to the pathway analysis partly addressed previous comments. However, the authors still didn't directly explain why the pathways from the combined data are more reasonable. One thing can be done is to look at the extra genes identified from combined data but not from single-platform data. If interesting pathways are enriched in those genes, it may make more sense.

Reviewer #2 (Remarks to the Author):

The revision addresses all the concerns I had. I have no further comments and believe the paper is now ready for publication.

Reviewer #4 (Remarks to the Author):

The authors addressed most of my questions, but I still have a concern about the performance of iPAGE, which is a strategy for data integration without preprocessing the expression data (<https://doi.org/10.1093/bioinformatics/btac379>, <https://doi.org/10.1093/bib/bbac002>, <https://doi.org/10.1016/j.compbiomed.2022.105881>). In other words, no normalization methods are required when using this method to select gene features. In comparison to the normalization methods used in this study, is iPAGE an effective cross-platform normalization that enables model training on array and RNA-seq data? At least, the authors should discuss the possible advantages and limitations about it.

Reviewer #1 (Remarks to the Author):

1. The authors mentioned in the last paragraph of the Introduction that the normalization approaches are applied to multiple applications: supervised machine learning, unsupervised machine learning, and pathway analysis. However, in the following sections, pathway analysis is discussed in the section on unsupervised machine learning. Therefore, pathway analysis should be part of unsupervised learning instead of a separate application. Furthermore, the author can explain more on the reason for focusing on those supervised and unsupervised machine learning applications.

We appreciate the reviewer's suggestion to provide more context for the machine learning applications we selected. We chose subtype and mutation status prediction for our supervised machine learning tasks because they are commonly used in cancer genomics studies and the labels are well-defined in our data. Our unsupervised machine learning results focus on pathway analysis since this is an important downstream application for understanding biologically-relevant gene expression patterns. We have added that text along with a clarification that pathway analysis is included as a type of unsupervised learning. Please see our updates at lines 61-67:

Here, we present a series of experiments to test what normalization approaches can be used to combine microarray and RNA-seq data for supervised machine learning and unsupervised machine learning, including pathway analysis. We chose subtype and mutation status prediction for our supervised machine learning tasks because they are commonly used in cancer genomics studies and the labels are well-defined in our data. Our unsupervised machine learning results focus on pathway analysis since this is an important downstream application for understanding biologically-relevant gene expression patterns.

2. In the Methods section, the authors introduced the LOG transformation. Specifically, "the microarray data was inverse log-transformed and then log-transformed...". Is inverse-transformation same as taking exponential? If so "inverse log-transformed and then log-transformed" does not make any change. The authors should give a clearer definition of the "inverse log-transform".

Thank you – we have clarified in the text that the LOG retransformation occurs after adding 1 to each expression value, forcing negative values to be non-negative. Please see our updated phrasing at lines 164-166:

Log2-transformation (LOG): As the log2-transformed array data contained negative values, the microarray data was inverse log-transformed and then log-transformed again after adding 1 to each expression value such that re-transformed values are non-negative.

3. In the introduction of Quantile normalization with CrossNorm, there is a typo in the sentence "we array and RNA-seq sample expression values were rescaled zero-to-one." It should be "the array and RNA-seq sample expression values ...".

Thank you! We have fixed this typo.

4. In the description of Figure 2 and 3, it says "violin plots...". However, the plots are not violin plots.

We appreciate the reviewer's close examination of the plot style in Figures 2 and 3. We agree that the violin plots may often appear to not be violin plots. This linear appearance was due to the range of values, the large point added at the median value, and the thickness of the outlines used in these plot elements. Based on the reviewer's experience, we have redesigned the plots to more clearly communicate the essential information – specifically the median Kappa values from each experimental setting and the line connecting the median values as the percentage of RNA-seq increases. In particular, we replaced the violin elements in favor of error bars around the median, which were calculated as $\pm 1.58 \cdot \text{IQR} / \sqrt{n}$, where IQR is the interquartile range, and n is the number of observations (10 repeats) (McGill R, Tukey JW, Larsen WA. Variations of Box Plots. Am Stat. 1978;32: 12–16.). We have updated the plots in Figures 2-3 and S3-S6 to reflect this change as well as the corresponding figure legends. Please see an example below represented by our updated Figure 2 and figure legend:

Figure 2. BRCA subtype classifier performance on microarray and RNA-seq test data. (A) Median Kappa statistics from 10 repeats of steps 1-3A from Figure 1 and for seven normalization methods (and untransformed data) are displayed. Median values are shown as points, and approximate 95% confidence intervals are shown around each median defined as $\pm 1.58 \cdot \text{IQR} / \sqrt{n}$ with IQR = interquartile range and n = number of observations.⁵⁹ (LOG - log2-transformed; NPN – nonparanormal normalization; QN – quantile normalization; QN (CN) – quantile normalization with CrossNorm; QN-Z – quantile normalization followed by z-score; TDM – Training Distribution Matching; UN – untransformed; Z – z-score)

5. The negative control added to the pathway analysis partly addressed previous comments. However, the authors still didn't directly explain why the pathways from the combined data are more reasonable. One thing can be done is to look at the extra genes identified from combined data but not from single-platform data. If interesting pathways are enriched in those genes, it may make more sense.

We thank the reviewer for this additional guidance and have now done more to analyze the pathways identified by PLIER. For each possible pathway, we determined how often that pathway was significantly associated (FDR < 0.05) with at least one latent variable across each of our data normalization conditions. We found that doubling the sample size led to important cancer pathways being identified more regularly in both the single platform and cross-platform settings, including pathways related to ERB2, NFKB immune infiltration, and RAF in breast cancer, and KRAS, MYC, and PRC2-related pathways in glioblastoma.

By focusing on pathways that are reliably detected more often in the full sample size data, we can identify stable patterns that emerge when more data is available. We ran PLIER 10 times using each combination of data normalization method and sample size (half or full). Out of those 10 runs, we found the proportion of times each oncogenic pathway was significantly associated with a latent variable. We set an arbitrary “meaningful difference” threshold of 0.2 to detect when that proportion was meaningfully greater using full sample size compared to half sample size and require array and RNA-seq data to both to satisfy the condition. We also arbitrarily limit results to those pathways found in over half of runs using the full size data.

We have included two new supplemental tables (Supplementary Tables 2-3) with the full set of pathway results for BRCA and GBM cancer types. We also now include an analysis notebook in our github repository (https://github.com/greenelab/RNAseq_titration_results) showing how we derived the additional gained pathways. Please also see the additional results reported in the main text (lines 332-335):

To evaluate the potential benefit of increasing sample size to pathway stability and consistency, we identified oncogenic pathways that were more frequently (change in proportion ≥ 0.2) associated with at least one latent variable in the full sample size data sets compared to the half sample size data sets.

And lines 436-441:

We found that doubling the sample size led to important cancer pathways being identified more stably and regularly in both the single platform and cross-platform settings, including pathways related to ERB2, NFKB immune infiltration, and RAF in BRCA, and KRAS, MYC, and PRC2-related pathways in GBM. Pathway results are included for BRCA (Supplementary Table 2) and GBM (Supplementary Table 3).

Reviewer #2 (Remarks to the Author):

The revision addresses all the concerns I had. I have no further comments and believe the paper is now ready for publication.

Thank you! We appreciate your time and effort improving this manuscript.

Reviewer #4 (Remarks to the Author):

The authors addressed most of my questions, but I still have a concern about the performance of iPAGE, which is a strategy for data integration without preprocessing the expression data (<https://doi.org/10.1093/bioinformatics/btac379>, <https://doi.org/10.1093/bib/bbac002>, <https://doi.org/10.1016/j.compbiomed.2022.105881>). In other words, no normalization methods are required when using this method to select gene features. In comparison to the normalization methods used in this study, is iPAGE an effective cross-platform normalization that enables model training on array and RNA-seq data? At least, the authors should discuss the possible advantages and limitations about it.

We thank the authors for their additional methods suggestion. Based on the reviewer's previous recommendation, we included CrossNorm (<http://dx.doi.org/10.1038/srep18898>) as a normalization method in our analysis. We could include CrossNorm because it fits within our experimental paradigm of testing normalization methods to shape RNA-seq data to match array data. The additional methods recommended here represent a different class of method employing pairwise gene-gene comparisons defined by the ratio between genes. The distinct nature of these methods makes them an interesting area for future work. We have described the advantages of this class of methods, including work from Ilya Shmulevich (2013, <https://doi.org/10.1038/nmeth.2445>) and the recommended citations showing recent application of this approach, in the discussion (lines 502-506):

Alternative approaches may not require reshaping RNA-seq data to match array data. For example, methods utilizing gene-pair ratios may rely on relative expression levels of genes within samples to identify useful features⁵⁵ and should be considered for future work, including recent approaches like iPAGE that apply this idea across platforms and data types.⁵⁶⁻⁵⁸

There were no reviewer comments in this round of revision. We again thank the reviewers for their time and effort to improve this work. Thank you!